# Artificial Intelligence Tools in Pediatric Urology: A Comprehensive Review of Recent Advances

**DOI:** 10.3390/diagnostics14182059

**Published:** 2024-09-17

**Authors:** Adiba Tabassum Chowdhury, Abdus Salam, Mansura Naznine, Da’ad Abdalla, Lauren Erdman, Muhammad E. H. Chowdhury, Tariq O. Abbas

**Affiliations:** 1Department of Electrical and Electronic Engineering, University of Dhaka, Dhaka 1000, Bangladesh; adibat500@gmail.com; 2Department of Electrical & Computer Engineering, Rajshahi University of Engineering & Technology, Rajshashi 6204, Bangladesh; salam35.ruet17@gmail.com; 3Department of Computer Science & Engineering, Rajshahi University of Engineering & Technology, Rajshashi 6204, Bangladesh; naznine31@gmail.com; 4Faculty of Medicine, University of Khartoum, Khartoum 11115, Sudan; 5James M. Anderson Center for Health Systems Excellence, Cincinnati, OH 45255, USA; larunerdman1@gmail.com; 6School of Medicine, University of Cincinnati, Cincinnati, OH 45267, USA; 7Electrical Engineering, Qatar University, Doha 2713, Qatar; mchowdhury@qu.edu.qa; 8Pediatric Urology Section, Sidra Medicine, Doha 26999, Qatar; 9College of Medicine, Qatar University, Doha 2713, Qatar; 10Weil Cornell Medicine Qatar, Doha 24144, Qatar

**Keywords:** artificial intelligence, artificial intelligence applications, diagnostic accuracy, medical imaging, pediatric medicine, predictive modeling, pediatric urology, surgical challenge

## Abstract

Artificial intelligence (AI) is providing novel answers to long-standing clinical problems, and it is quickly changing pediatric urology. This thorough analysis focuses on current developments in AI technologies that improve pediatric urology diagnosis, treatment planning, and surgery results. Deep learning algorithms help detect problems with previously unheard-of precision in disorders including hydronephrosis, pyeloplasty, and vesicoureteral reflux, where AI-powered prediction models have demonstrated promising outcomes in boosting diagnostic accuracy. AI-enhanced image processing methods have significantly improved the quality and interpretation of medical images. Examples of these methods are deep-learning-based segmentation and contrast limited adaptive histogram equalization (CLAHE). These methods guarantee higher precision in the identification and classification of pediatric urological disorders, and AI-driven ground truth construction approaches aid in the standardization of and improvement in training data, resulting in more resilient and consistent segmentation models. AI is being used for surgical support as well. AI-assisted navigation devices help with difficult operations like pyeloplasty by decreasing complications and increasing surgical accuracy. AI also helps with long-term patient monitoring, predictive analytics, and customized treatment strategies, all of which improve results for younger patients. However, there are practical, ethical, and legal issues with AI integration in pediatric urology that need to be carefully navigated. To close knowledge gaps, more investigation is required, especially in the areas of AI-driven surgical methods and standardized ground truth datasets for pediatric radiologic image segmentation. In the end, AI has the potential to completely transform pediatric urology by enhancing patient care, increasing the effectiveness of treatments, and spurring more advancements in this exciting area.

## 1. Introduction

The field of pediatric urology is changing as a result of artificial intelligence (AI) applications, which are providing novel answers to persistent problems with patient diagnosis and therapy. This paper examines the role of AI in pediatric urology, offering a thorough examination of implications, current uses, and future potential to improve standards of care.

### 1.1. Background on Pediatric Urology

Pediatric urology focuses on the diagnosis and management of urinary tract problems in children and adolescents, which can include a broad range of conditions such as urologic cancers, voiding dysfunction, urinary tract infections, and complex congenital defects. A multidisciplinary strategy that combines pediatricians, urologists, nephrologists, radiologists, and other allied healthcare specialists is typically necessary for the treatment of pediatric urological problems [1]. Depending on the exact illness and severity, several treatment plans may be used, with possible interventions ranging from medication to surgery.

Diagnosis and management of juvenile urological disorders have regularly been transformed by technological progress, including minimally invasive surgery and sophisticated imaging techniques. However, this discipline still experiences a number of major challenges due to the intricacy and diversity of certain conditions, the requirement for pediatric-patient-specific techniques, and the need for long-term follow-up to monitor complications and ensure optimal results. Complexity is further compounded by the variation in diseases and how they appear across pediatric patients, since ailments can present in different ways depending on age, genetics, and comorbidities. Furthermore, due to the dynamic nature of childhood development, treatment plans must be continuously modified to account for changes in growth and development [2]. In recent years, the use of AI in pediatric urology has shown promise for improving patient diagnosis, treatment planning, surgical results, and postoperative care. The purpose of this paper is to present a thorough analysis of recent progress in AI applications for pediatric urology, emphasizing the many advantages, current difficulties, and future possibilities in this quickly developing area.

### 1.2. Importance of AI in Pediatric Urology

AI has the potential to reshape the field of pediatric urology by delivering better clinical judgment, increasing diagnostic precision, and improving patient outcomes. The special challenges when treating young patients make the use of AI in pediatric urology even more important. Children frequently exhibit intricate and varied urological ailments that require customized methods for precise identification and efficient therapy, as shown in Figure 1.

The ability of AI to quickly and accurately assess enormous volumes of clinical and imaging data is one of the technology’s main advantages. By sorting through complex patterns and nuanced details in patient information, machine learning (ML) algorithms can aid doctors with early detection, risk assessment, and individualized therapy planning. In pediatric populations, where prompt intervention can have a substantial influence on quality of life and long-term outcomes, this capacity is extremely valuable [3] (Table 1).

By synthesizing information from many sources, AI-powered decision support systems provide healthcare providers with crucial evidence-based advice. These technologies can assist pediatric urologists in navigating intricate treatment algorithms, optimizing surgical planning, and reducing intervention-related risks to ultimately improve results for pediatric urology patients [4]. In particular, healthcare professionals can now employ AI tools to provide more individualized, accurate, and efficient care.

With enhanced tools for image interpretation, segmentation, and visualization, AI has the potential to completely transform medical imaging in pediatric urology. Deep learning (DL) algorithms can evaluate radiological images with unprecedented accuracy, making it easier to identify tiny anomalies and direct focused interventions. Furthermore, AI-driven image analysis methods make it possible to assess the clinical course of a disease, response to medication, and impact of treatment, providing clinicians with important new data on patient care [1].
diagnostics-14-02059-t001_Table 1Table 1Role of AI in enhancing pediatric urology.Aspect Importance of AI in Pediatric Urology Diagnosis AI can aid in the accurate and early diagnosis of pediatric urological conditions through analysis of medical images and patient data [5]. Treatment Planning AI algorithms can assist in developing personalized treatment plans based on patient-specific factors, improving outcomes, and reducing risks [6]. Surgical Assistance AI-enabled surgical tools can enhance precision and safety during pediatric urological procedures, reducing complications and recovery times [7]. Predictive Analytics AI can help predict progression of certain urological conditions in pediatric patients, allowing for proactive intervention and management [3]. Research and Innovation AI facilitates the analysis of large datasets to identify trends, patterns, and novel insights, driving advances in pediatric urology [8]. Education and Training AI-powered simulations and virtual reality environments offer valuable educational resources for training pediatric urologists and residents [8]. Patient Monitoring AI-driven monitoring systems can continuously track pediatric urology patients, providing real-time alerts for any concerning developments [6]. 


Beyond diagnosis and therapy, AI is also transforming pediatric urology research and innovation. Enormous datasets can now be evaluated using ML techniques to discern new information on prognostic variables, therapy responses, and disease causes. AI-powered virtual reality simulations are also revolutionizing surgical education by giving pediatric urology trainees hands-on experience in a safe setting. AI will therefore make pediatric urology more precise, efficient, and patient-centered in the future by accelerating scientific innovation and discovery [3].

### 1.3. AI Applications in Other Medical Fields

AI techniques have been increasingly applied across different medical fields to enhance diagnostic and predictive capabilities. For instance, Ashrafi et al. (2024) [9] employed a deep learning model to predict mortality among patients on mechanical ventilation in the ICU, comparing its performance with that of seven other machine learning models. Analyzing the data from 16,499 patients, the deep learning model achieved an accuracy of 85.9% and an AUC of 87.9%. It identified respiratory failure and the duration of ventilation as significant factors influencing mortality, offering valuable insights for patient care and outcome prediction in the ICU.

Similarly, Pishgar et al. (2021) [10] used a mining-based approach to predict mortality in 46,476 patients with paralytic ileus (PI) within 24 h of admission. Their process mining model for PI mortality (PMPI) achieved an AUC of 0.82, accurately predicting the death of 53 patients, while misclassifying 3. SHAP analysis revealed that demographic factors had the most significant impact, underscoring the importance of medical history in evaluating PI risk.

Theis et al. (2020) [11] developed several machine learning models, including support vector machines (SVMs), k-nearest neighbors, decision trees, linear discriminant analysis, and Gaussian naïve Bayes, to forecast mortality. The proposed mining/deep learning architecture integrated the medical histories of people with diabetes with severity scores, with the SVM model achieving the highest AUC of 81.38%.

Bodaghi et al. (2024) [12] created an efficient intermediate fusion network that leverages manifold learning to detect stress through biometric signals such as heart rate, electrodermal activity, skin temperature, and acceleration data. The study highlighted the importance of data balancing through downsampling, which improved dimensionality reduction and model training. Adding a 1D CNN layer led to a 7.41% increase in accuracy, a 6.18% boost in precision, a 9.23% improvement in recall, and a 9.42% enhancement in the F1-score.

Duff (2023) [13] developed an automated pipeline using radiomic biomarkers from [18F]-fluorodeoxyglucose positron emission tomography-computerized tomography (FDG PET-CT) images to diagnose active aortitis. The convolutional neural network (CNN) segmented the aorta from the FDG PET-CT images of patients with aortitis and controls, achieving AUC values greater than 0.8. These promising results suggest that this radiomic pipeline could be generalized and potentially integrated into an automated clinical decision support system for consistent and unbiased evaluations.

### 1.4. Objectives of This Review

This review aims to shed light on the potential of AI to drive revolutionary advances in key pediatric urological disorders and surgeries (Figure 2), while navigating the ethical, regulatory, and practical challenges of adopting new technologies:

Examining the technical landscape: Our initial goal was to present a broad overview of the technical environment where pediatric urology and artificial intelligence converge. We aimed to summarize the current research and recent breakthroughs to provide readers with a thorough grasp of AI tools and approaches that are changing clinical practice in this specialized sector.

Clinical implications: Beyond the technical details, we aimed to clarify the real-world effects of adopting AI in pediatric urology, both advantages and pitfalls. We explored how AI-driven innovations are revolutionizing diagnostic routes, treatment paradigms, and patient outcomes through a critical examination of current research and case studies.

Regulatory and ethical boundaries: Mindful of the bioethics and legal issues that accompany technological advances in healthcare, we endeavored to traverse these boundaries, identify ethical concerns, highlight privacy issues, and outline legal frameworks that will influence incorporation of AI into pediatric urology.

Knowledge gaps and future directions: Identifying knowledge gaps and potential topics for additional research into AI applications for pediatric urology was a major goal of this review. Through a critical assessment of the current literature, we aimed to identify key areas for future AI-driven research initiatives and innovations.

Encouraging medical professionals and scholars: Our main goal was to provide healthcare professionals, researchers, and other stakeholders with the information and understanding necessary to fully realize AI’s transformational potential in the field of pediatric urology. Our aim was to present a range of evidence-based discoveries and viewpoints that stimulate meaningful debate, cooperation, and innovation in this dynamic and quickly changing sector.

## 2. Hydronephrosis

Hydronephrosis is a medical condition that involves the enlargement and distension of one or both kidneys due to the accumulation of urine (Figure 3). Table 2 shows the challenges of managing hydronephrosis in pediatric urology, including the need for precise diagnosis and customized therapeutic approaches. Historically, the evaluation of hydronephrosis has relied on subjective grading systems and intrusive diagnostic procedures, resulting in inconsistency in medical practice and possible hazards for patients. However, AI tools now offer promising alternatives to boost the precision of diagnoses, simplify clinical processes, and improve patient results. According to the Society for Fetal Urology (SFU) grading system, ultrasound scans can be used to determine hydronephrosis severity from Grade 1 to 4. In Figure 4, the arrows show how a kidney with hydronephrosis differs from a normal kidney in terms of urine flow. The urine flows freely and smoothly from the renal pelvis via the ureter and into the bladder in a normal kidney (left). On the other hand, the arrow in the hydronephrotic kidney (right) denotes a poor urine flow brought on by a kidney stone-related blockage of the ureter. This obstruction causes hydronephrosis, or enlargement of the renal pelvis where urine collects and causes kidney distention.
Figure 3Anatomy and urine flow in healthy versus hydronephrosis kidneys [14].
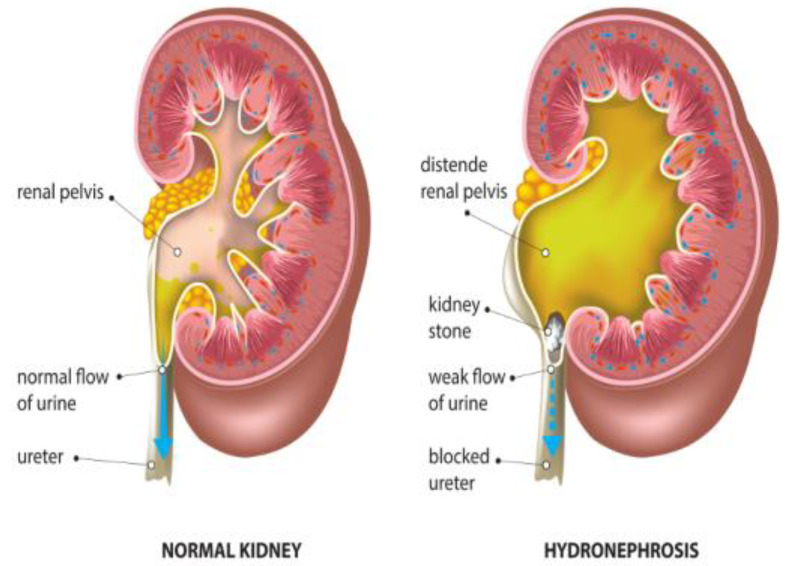


### 2.1. Diagnostic Challenges in Pediatric Hydronephrosis

The traditional methods of diagnosing pediatric hydronephrosis are limited by the subjective nature of ultrasound imaging and the variability in interpretation [15]. Semiquantitative ultrasonic analysis can result in observer-dependent outcomes, with a lack of explicit instructions for subsequent monitoring and management. In addition, establishing precise criteria for intervention based on grading methods such as the SFU [16] and the Urinary Tract Dilatation (UTD) [17] classification systems remains challenging. This ambiguity increases the need for invasive treatments, radiation exposure, and frequent testing, which place additional strain on young patients, their families, and healthcare systems [18].
Figure 4Kidney ultrasound scans indicating different levels of hydronephrosis: (**a**) Grade 1, (**b**) Grade 2, (**c**) Grade 3, and (**d**) Grade 4. Based on SFU grading system [19].
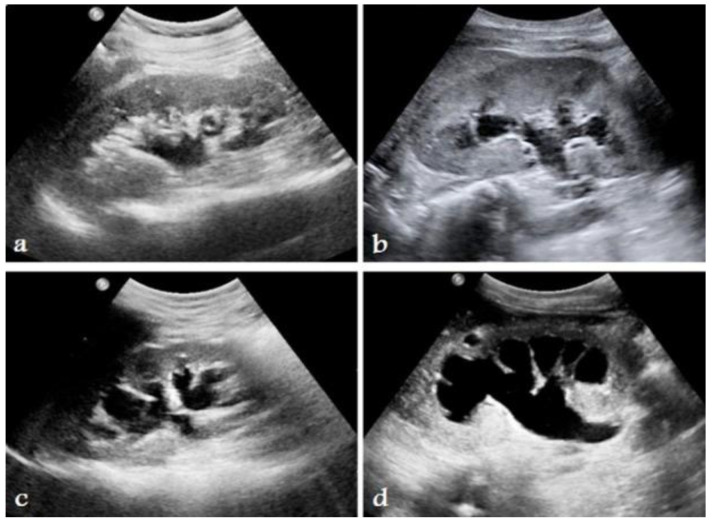



The constraints of conventional approaches also apply to imaging techniques such as abdominal CT, which is highly efficient but also raises issues regarding radiation exposure and cost. While ultrasound is easily accessible, this method is also hindered by interobserver variability, which compromises reliability [20]. The lack of clear guidelines and understanding of hydronephrosis natural history further complicates decision making, thus impacting patient counseling and management [21].

DL models may help address these difficulties by offering more impartial and replicable diagnostic tools [22]. These models can reduce the subjective, inconsistent, and variable nature of existing diagnostics, although further work will be required to maximize the accuracy of these approaches, particularly for the grading of severe cases [23].
diagnostics-14-02059-t002_Table 2Table 2Diagnostic challenges in pediatric hydronephrosis.Diagnostic Challenge Description Mitigation Strategies Subjective nature of ultrasound imaging [24]Traditional methods for diagnosing pediatric hydronephrosis rely heavily on ultrasound imaging, which is subjective and prone to variability. Implementing DL models can offer more impartial and replicable diagnostic tools, reducing the subjective and variable nature of ultrasound interpretation. Lack of clear guidelines for intervention [25]Establishing precise criteria for intervention based on grading systems presents challenges. Continued research is needed to develop clearer guidelines for intervention, considering factors such as patient age, severity of hydronephrosis, and potential risks. Issues with imaging techniques [20]Although imaging techniques like abdominal CT are efficient, they pose risks of radiation exposure and higher cost. Exploring alternative imaging modalities with lower radiation exposure, such as magnetic resonance urography (MRU), can mitigate the risks associated with radiation exposure while maintaining diagnostic efficiency. 

### 2.2. AL Solutions for Hydronephrosis Management

Quantitative imaging and ML methods can now be used to assess renal sonograms with the goal of predicting the need for diuretic nuclear renography [10]. By analyzing key morphological features, this type of approach was successfully used to determine ultrasound-based safety criteria that correspond to washout times, potentially leading to a 62% reduction in the use of invasive diagnostic procedures. In addition, researchers created DL algorithms that can accurately identify obstructive hydronephrosis using only ultrasound scans, which could speed up this process and significantly improve clinical care [18].

AI has demonstrated potential in the management and grading of hydronephrosis, providing more consistent and accurate evaluations. Hydronephrosis is graded based on renal pelvis dilation—a critical component in the diagnosis of urinary tract obstructions [26]. The ability of AI to differentiate between hydronephrosis resulting from posterior urethral valve (PUV) and other etiologies was demonstrated in research conducted by Ostrowski et al. (2023) [27]. Their research showed how AI might improve diagnosis accuracy, resulting in better patient outcomes through prompt and targeted therapies.

Advanced DL models such as U-Net, Res-UNet, and UNet++ have also been used to identify hydronephrosis from ultrasound images, aiming to improve efficiency and uniformity in clinical environments [20]. The combination of these models, along with pre-processing and postprocessing techniques, exhibited a remarkable level of accuracy in identifying moderate/severe hydronephrosis. ML algorithms were also used to distinguish between high- and low-grade hydronephrosis to facilitate standardized automatic grading These algorithms have been used to create predictive models using a combination of radiomic texture analysis, support vector machines, and DL approaches. In addition, DL algorithms have been used to segment ultrasound pictures and calculate the hydronephrotic area as a ratio of the renal parenchyma, which serves as a useful biomarker of disease severity [25]. Overall, AI technologies offer higher diagnostic accuracy, clinical efficiency, and improved care of hydronephrosis management.

### 2.3. Impact on Treatment Planning and Long-Term Monitoring

AI approaches have significant implications for treatment planning and long-term monitoring of pediatric hydronephrosis. Cerrolaza et al. (2016) [15] showcased the ability of AI algorithms to accurately forecast the need for diuretic renography. This approach provides a noninvasive means of evaluating the extent of hydronephrotic renal units. Clinicians can optimize treatment planning and reduce unnecessary invasive procedures by determining thresholds of clinically meaningful washout times. This will allow the efficient selection of patients who would benefit from diuretic renography, thereby decreasing radiation exposure and the associated healthcare costs. This technique is particularly relevant to the continuing debate over whether postnatal imaging is necessary in newborns with prenatally diagnosed hydronephrosis.

Erdman et al. (2020) [18] revolutionized treatment planning in pediatric urology by introducing DL algorithms that can predict obstructive hydronephrosis from ultrasound pictures alone. The exceptional accuracy attained by these models allows medical professionals to identify patients at high risk who require urgent surgical intervention, thereby minimizing the need for unnecessary invasive testing and follow up. Implementation of standardized management protocols across all urological departments also ensures uniform and precise identification of hydronephrosis, particularly in remote regions lacking specialized pediatric facilities.

In addition, Lien et al. (2023) [20] highlighted the importance of identifying hydronephrosis promptly and correctly, particularly in emergency and acute care environments. AI algorithms can precisely identify moderate/severe hydronephrosis to speed up medical treatment and potentially decrease the duration of emergency room visits. The implementation of automated hydronephrosis grading from ultrasound scans also improves diagnostic methods and standardizes evaluation, facilitating the customization of treatment plans for individual patients.

Sloan et al. (2023) [21] introduced AI systems that can forecast clinical outcomes by analyzing ultrasound images collected over an extended period. This technique offers insight into the evolution of hydronephrosis, the success of different treatments, and ongoing patient management. In addition, the HARP ratio can be calculated using DL techniques, as explained by Song et al. (2022) [23]. This AI technique streamlines the calculation process and reduces inconsistencies among observers, enabling the precise and consistent monitoring of hydronephrosis progression to ensure reliable long-term care.

Overall, the incorporation of AI methods into the management of pediatric hydronephrosis shows great promise to improve treatment planning and long-term monitoring tactics. AI-driven solutions in pediatric urology provide more accurate, effective, and tailored approaches to patient treatment by predicting the need for diagnostic procedures, automating severity grading, and tracking disease development.

## 3. Pyeloplasty

A cornerstone of surgical care for pediatric urological disorders is pyeloplasty, a surgical operation designed to correct blockage of the ureteropelvic junction (UPJ), as shown in Figure 5. Despite historically high rates of success, pediatric pyeloplasty poses specific difficulties because of the tiny stature and fragile anatomy of young patients. The complex surgical challenges of pediatric pyeloplasty, the development of AI-assisted operating methods, and potential future applications are outlined in this section (Kelley et al., 2016) [28].
Figure 5Normal and obstructed ureteropelvic junction (Ariyanagam, 2024) [29].
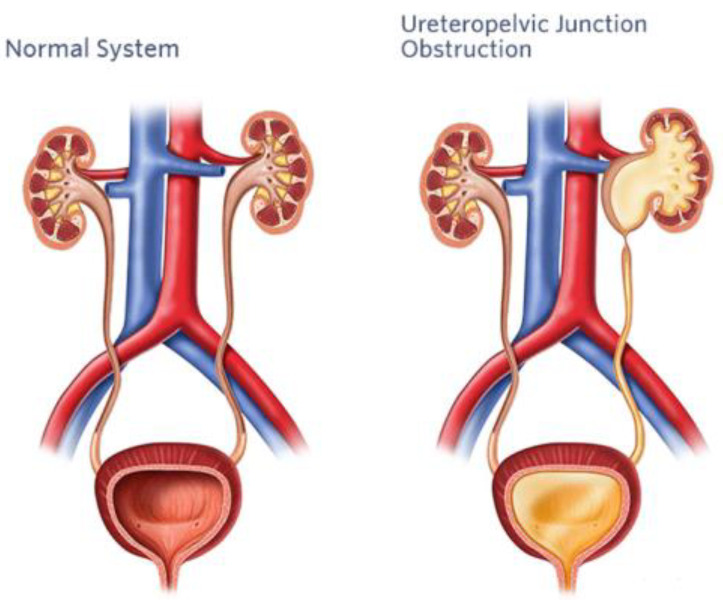



### 3.1. Surgical Challenges in Pediatric Pyeloplasty

Although pediatric pyeloplasty is a fundamental surgical procedure for treating blockage of the UPJ, the fragile anatomy of pediatric patients requires extreme accuracy and attention to detail. A primary obstacle is the restricted area available for surgery and the requirement for careful tissue manipulation to maximize results while reducing harm [30]. The diverse appearances of UPJ blockages also complicate surgical planning and management. Individualized surgical techniques may be required for pediatric patients due to a variety of anatomical variables, including crossing vessels, aberrant anatomy, and related congenital defects. To successfully address these factors, one must have a solid grasp of pediatric anatomy and pathology in addition to a flexible surgical approach [29,30,31].

The management of postoperative complications in pediatric pyeloplasty remains complex [32]. Urine leakage, blockage recurrence, and renal parenchymal loss are example complications that need to be identified and treated promptly to minimize side effects and maximize long-term renal function. In order to reduce morbidity and maintain renal health, surgical treatments must also be carefully considered and strategically planned in the light of possible effects on future renal growth and development [33]. Structural and physiological variations in pediatric patients create additional surgical hurdles for pyeloplasty. During these procedures, a number of major issues have been identified by Esposito et al. (2023) [34] and Masieri et al. (2020) [35], along with possible mitigation techniques:

Small anatomy: The smaller anatomical components of pediatric patients make precision surgical operations more difficult. Surgeons use pediatric surgical procedures that are adapted to the reduced scale of pediatric anatomy in conjunction with tiny tools to solve this challenge.

Reduced vision: Vision is typically reduced for pediatric patients during surgery due to the small size of the operative region. Surgeons use magnification techniques such as surgical loupes or endoscopic magnification equipment to increase visibility. This allows the clearer visualization of fine anatomical features within the operative site.

Tissue handling: Delicate tissues from pediatric patients must be handled with extreme caution to avoid harm and ensure positive results. To reduce tissue stress and improve outcomes, surgeons prioritize precise hemostasis, use thin surgical equipment intended for pediatric operations, and manipulate tissue gently.

Ensuring the safety and efficacy of pediatric pyeloplasty requires attention to detail and specific strategies to address surgical obstacles. Nonetheless, pediatric pyeloplasty remains a very popular and successful surgical procedure for treating UPJ blockage in young children.

### 3.2. AI-Assisted Surgical Techniques

AI is transforming pediatric urology by providing new methods and instruments to improve the accuracy, security, and effectiveness of major surgical procedures. Preoperative planning and simulation are two of the main ways in which AI have been deployed in pediatric pyeloplasty. AI algorithms can analyze preoperative images including CT, MRI, and ultrasound to produce three-dimensional reconstructions of the urinary system and determine the optimal surgical technique. AI-driven preoperative planning technologies also allow surgeons to customize treatment plans based on the distinct anatomy of each pediatric patient by giving them access to comprehensive anatomical data in virtual simulations [36].

Intraoperative surgical navigation systems improve accuracy and precision by combining AI algorithms with real-time imaging data to give surgeons enhanced visibility and guide the surgical process. AI-assisted or robotic navigation systems make precise dissection, tissue manipulation, and suture insertion easier by superimposing anatomical landmarks, vascular maps, and trajectory planning onto the surgical field. This reduces error and improves surgical results [36]. AI systems can identify minute variations in surgical dynamics and notify surgeons of possible consequences or departures from best practices by evaluating intraoperative data, such as motion of surgical instruments, tissue properties, and physiological factors. Proactive advising improves situational awareness and makes it possible to take prompt action to reduce risks and improve patient outcomes [37].

AI-driven surgical education and training systems are also revolutionizing the way pediatric urology trainees learn and hone their surgical skills outside of the operating room. AI-powered virtual reality simulations give students rich, interactive settings in which to refine their procedural abilities, practice advanced surgical methods, and model intricate surgical situations. AI-assisted surgical training platforms also help pediatric urology trainees gain proficiency and confidence in executing pyeloplasty operations by providing a secure and regulated learning environment [38].

### 3.3. Outcomes and Future Directions

Offering excellent long-term results and high success rates, pediatric pyeloplasty is a mainstay in the surgical treatment of UPJ blockage. Improvements in perioperative care, follow-up procedures, and surgical methods have led to outstanding results in juvenile pyeloplasty, including the remission of symptoms, preservation of renal function, and avoidance of sequelae [33].

Studies that track patients over an extended period of time have shown the durability of surgical results after juvenile pyeloplasty; most patients report long-lasting symptom alleviation and the maintenance of renal function. Furthermore, advances in less-invasive surgical methods, such laparoscopic and robotic-assisted procedures, have improved outcomes further by decreasing surgical morbidity, improving cosmesis, and hastening recovery [34].

AI has the potential to significantly impact the future of pediatric pyeloplasty by providing personalized treatment recommendations catered to the specific requirements of each patient, real-time intraoperative guidance, and predictive modeling for risk stratification. By using cutting-edge imaging modalities, predictive analytics, and ML algorithms, AI-driven surgical platforms and decision support systems can increase surgical accuracy, improve outcomes, and minimize complications. AI-powered virtual reality training programs will also encourage professional growth in the field of pediatric pyeloplasty that will positively impact treated patients [35]. At the same, a number of ethical, regulatory, and practical challenges must be carefully addressed to ensure the appropriate deployment of AI in pediatric surgical settings [34].

## 4. Pyeloplasty: Kidney Tumors and Stones

Pediatric kidney tumors and stone care call for a multidisciplinary approach to ensure accurate diagnosis and efficient treatment. In this section, we explore the diagnostic nuances and treatment issues related to kidney cancers and stones, then explore the revolutionary effects of AI applications on patient management and outcomes.

### 4.1. Outcomes and Future Directions: Diagnostic Challenges and Treatment Options

Given the intricacy of kidney tumors and stones, as well as the distinct physiological features of juvenile anatomy, diagnosing and treating these illnesses in young patients present considerable obstacles:Diagnostic difficulties: Children with kidney tumors and stones may exhibit nonspecific symptoms such hematuria, stomach discomfort, or urinary tract infections, which can be mistaken for other common pediatric ailments. Therefore, a combination of clinical evaluation, imaging exams, and laboratory investigations is crucial for accurate diagnosis. However, it can be difficult to differentiate between different forms of stones or distinguish between benign and malignant tumors, requiring a thorough diagnostic approach [39].Imaging modalities: A number of imaging modalities, such as intravenous pyelography (IVP), CT, MRI, and ultrasound, are essential for the diagnosis of juvenile kidney cancers and stones. Ultrasound is noninvasive and emits no ionizing radiation, so this method is used as a first-line imaging modality to examine renal morphology and identify structural problems (Figure 6). CT and MRI are more sensitive and specific for assessing stone load and identifying renal masses but also entail radiation exposure and require patient sedation [40].Treatment options: Tumor type, size, and location, and patient age are among the characteristics that influence how juvenile kidney tumors and stones are managed. Options for treatment vary from minimally invasive techniques and surgical intervention to cautious maintenance and attentive waiting. The primary therapy for kidney cancers that are localized is surgical excision; nephron-sparing techniques are recommended whenever possible in order to maintain renal function. Alternatively, depending on the size and composition of the stones, medicinal therapy, dietary changes, and minimally invasive techniques like ureteroscopy or shock wave lithotripsy (SWL) may be employed [41].
Figure 6Unenhanced abdominal CT images (kidney: yellow; kidney stone: green) [42].
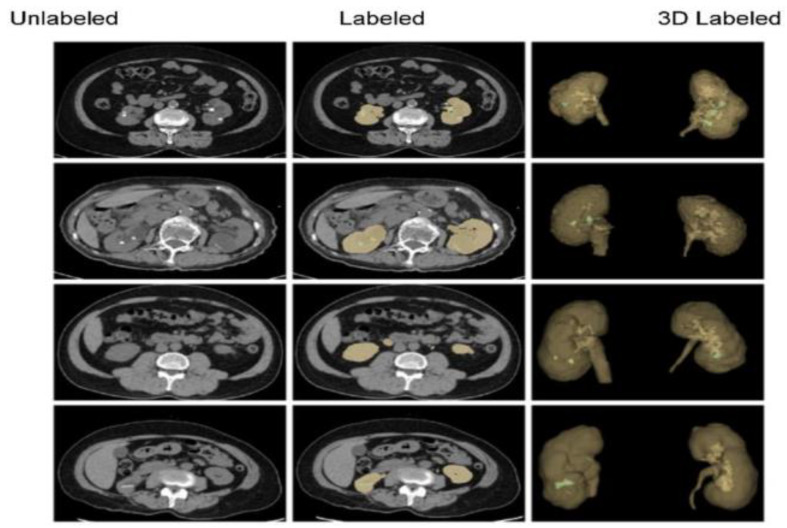



Clinicians should take into account the individual requirements and preferences of each patient when navigating the diagnostic hurdles and treatment options for kidney tumors and stones in children. Pediatric patients with kidney tumors and stones can receive better care and superior results can be achieved when practitioners use a multidisciplinary team and combine cutting-edge imaging methods with evidence-based therapy algorithms.

### 4.2. AI Applications in the Management of Kidney Tumors and Stones

AI technologies are transforming the way young patients with kidney tumors and stones are managed, in particular by guiding diagnosis and treatment planning to raise standards of care.

#### 4.2.1. Pediatric Kidney Stones

Improved diagnostics: AI-driven image analysis tools are improving the identification and measurement of juvenile kidney stones, enabling more precise diagnosis and therapy. Clinicians can estimate stone load and composition by applying ML techniques, making it possible to customize treatment and patient management [40].Personalized therapy planning: AI systems create individualized therapy suggestions based on the analysis of patient-specific information, including imaging data and medical history. This improves outcomes and lowers the likelihood of recurrence [39].Predictive modeling and risk stratification: Physicians can identify juvenile patients with kidney stones who are more likely to experience problems or a recurrence by using AI-powered predictive modeling. ML algorithms can forecast future stone occurrences and direct treatment and preventative actions by evaluating a variety of clinical factors and imaging data to achieve better long-term results [41].Ethics and regulations: Careful consideration of ethical and regulatory problems is required when integrating AI into the treatment of juvenile kidney stones, which should employ defined protocols and encourage openness and accountability as standard. For AI to be used ethically and responsibly, cooperation among researchers, regulatory agencies, and healthcare practitioners will be crucial [39].

#### 4.2.2. Pediatric Kidney Tumors

Improved diagnostics: AI-driven image analysis tools can enhance the identification and description of pediatric kidney cancers and tumors. Radiologists can spot minor abnormalities, characterize renal masses, and estimate tumor burden more accurately using ML algorithms that have been trained on large datasets of pediatric renal images (Figure 7). AI algorithms enable the early identification of kidney cancers and tumors by methods including pattern recognition, masking, segmentation, and quantitative analysis, allowing for confident diagnosis and timely treatment [24].Personalized treatment planning: AI systems provide personalized therapy recommendations and perform the prognostic evaluation of patient-specific data, including clinical history, imaging results, and laboratory values. For young patients with kidney cancers, this customized strategy enables medical practitioners to maximize therapeutic efficacy, reduce treatment-related morbidity, and enhance long-term outcomes [43].Predictive modeling and risk stratification: AI predictive modeling enables doctors to identify children with kidney tumors who are more likely to experience treatment failure or disease progression. ML algorithms identify prognostic characteristics and biomarkers through the analysis of varied datasets. This improves treatment outcomes and survival rates for children with kidney cancers by empowering doctors to use tailored risk mitigation strategies and start treatment early [44].Ethical and regulatory considerations: Incorporating AI into the treatment of juvenile kidney cancers and tumors requires careful consideration to overcome legal, ethical, and practical challenges. Collaboration between developers, healthcare providers, and other stakeholders will be essential for ensuring the ethical and effective use of AI in clinical practice [45].

**Figure 7 diagnostics-14-02059-f007:**
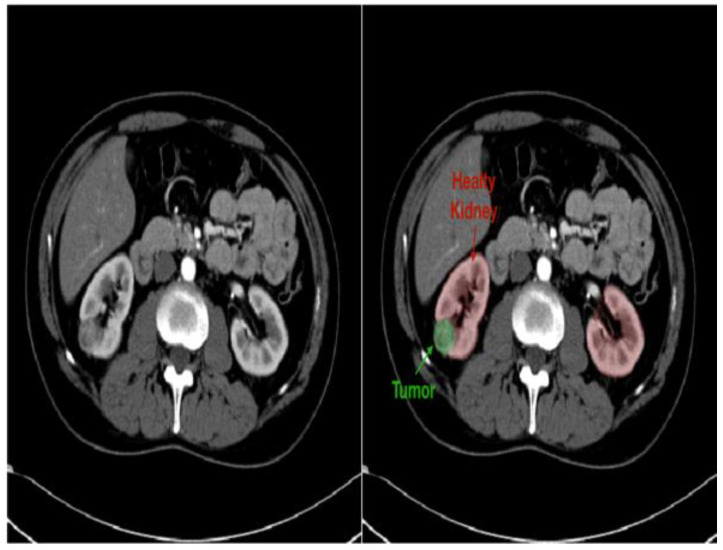
Kidney CT image in grayscale (**left**) and with overlaid mask (**right**) to highlight suspected stones or tumors [46].

### 4.3. Outcomes and Future Directions: Impact on Treatment Strategies and Outcomes

Treatment plans and patient outcomes will be significantly impacted by the use of AI in the management of kidney tumors and stones in young patients (Table 3).
diagnostics-14-02059-t003_Table 3Table 3AI treatment strategies and outcomes for pediatric kidney tumors and stones.Pediatric Kidney Care Description Example Importance Optimized treatment planning [44]AI-driven algorithms optimize treatment plans for kidney tumors and stones in children by evaluating vast databases of patient-specific data. Individualized treatment regimens tailored to each patient’s requirements based on clinical features, imaging results, and therapy responses. Maximizes therapeutic efficacy, reduces treatment-related morbidity, and enhances long-term results. Enhanced surgical precision [40]AI technologies provide greater precision and accuracy in surgical interventions for pediatric kidney tumors and stones. AI-powered surgical navigation systems offer immediate feedback during operations, facilitating accurate tumor and stone localization, ideal tissue resection margins, and shorter operating times. Improves surgical outcomes and patient safety by increasing precision and reducing intraoperative complications. Improved prognostic approaches [40] AI-powered predictive analytics revolutionize prognostication and risk stratification for pediatric kidney tumors and stones. ML algorithms analyze multidimensional datasets to uncover prognostic biomarkers, treatment response predictors, and disease progression indicators. Informs decisions about patient care and therapy selection, leading to enhanced treatment outcomes, reduced complications, and improved survival rates. Longitudinal monitoring and follow-up [41] AI technologies enable longitudinal monitoring and follow-up for pediatric patients with kidney tumors and stones beyond treatment planning and intervention. AI-powered monitoring systems examine long-term patient data to detect subtle changes in disease state, early indicators of recurrence or progression, and prompt interventions or treatment regimen modifications. Enhances disease monitoring, treatment adherence, and long-term outcomes by enabling proactive management and personalized follow-up care. 

## 5. Vesicoureteral Reflux

VUR is a common pediatric urological condition characterized by the retrograde flow of urine from the bladder into the ureters and kidneys. Failing to treat this condition can result in renal scarring and chronic kidney damage, highlighting the need for precise diagnosis and efficient management. Previously, the diagnosis of VUR depended on invasive techniques such as VCUG and radionuclide cystography, which cause discomfort and pose significant risks for pediatric patients. Figure 8 illustrates the VUR grading system, which classifies this disorder into five groups based on the extent of urine backflow and dilation of urinary tract structures.

Advances in medical imaging and AI have offered new possibilities for enhancing the diagnosis and treatment of VUR. Microwave radiometry provides a nonintrusive method for identifying VUR by monitoring temperature changes in the kidney as a result of urine reflux [47].

ML methods have also been used to improve the grading and diagnosis of VUR based on the automated analysis of features collected from imaging studies [48,49]. This section aims to explore the diverse array of methods currently used for the diagnosis and management of VUR in pediatric patients: Grade I: urine backflow into the ureter but not the kidney; Grade II: urine reaches the kidney but no renal pelvis dilation; Grade III: mild to moderate ureter and renal pelvis dilation; Grade IV: dilation of ureter, renal pelvis, and calyces; Grade V: severe dilation of ureters, renal pelvis, and calyces [50].

### 5.1. Diagnosis and Management of VUR

The conventional diagnosis and treatment of VUR are limited by their invasive nature. VCUG is the current gold standard method but requires catheterization and exposes patients to radiation. While commonly used, VCUG exhibits significant variability between observers due to the reliance on visual cues to grade severity, which can result in uneven grading and inconsistent decision making [48].

To overcome the limitations of VCUG, researchers have investigated alternate noninvasive methods. Voiding urosonography (VUS) with contrast agents has demonstrated enhanced sensitivity in identifying VUR compared with earlier contrast-based methods. Despite their excellent safety record, there are still concerns over the invasiveness of applying contrast agents. It is therefore crucial to continue searching for noninvasive techniques that offer high sensitivity and specificity for the diagnosis of VUR [47].

At present, diagnosis is heavily dependent on VCUG and the grading system established by the International Reflux Committee, which results in the subjectivity of and variability in image interpretation. Inconsistencies in VUR grading impact diagnostic accuracy, particularly when differentiating between many classes, underscoring the need for a more objective categorization method [49]. Further progress in imaging technologies is also needed to aid with the detection of small lesions and structural disorders [51]. New approaches such as contrast-enhanced voiding ultrasonography (ceVUS) may offer alternatives to conventional methods [52]. CeVUS is radiation-free and dependable for diagnosing VUR, surpassing VCUG in terms of sensitivity and specificity. This method effectively addresses concerns regarding radiation exposure and invasiveness. Nevertheless, it is imperative to tackle issues such as reliance on operators and inaccurate negative results by conducting additional research [53]. Ultimately, while traditional diagnostic approaches remain useful, continuous progress in noninvasive techniques and the creation of objective categorization systems are warranted to enhance the precision and uniformity of VUR diagnosis and treatment.

### 5.2. AI-Based Imaging Techniques

AI-based imaging tools represent a major breakthrough in the diagnosis and treatment of VUR. One proposed method entails the use of ML models to automate the assessment of VUR severity from VCUG data [48]. This strategy trains models using annotated images that have been assessed by multiple experts. Using ML techniques, this approach presents a possible transition toward more dependable and uniform diagnostic procedures in the management of VUR.

Quantitative vesicoureteral reflux (qVUR) is a novel method that uses supervised ML to assess the severity of VUR by analyzing VCUG image features [49]. The use of a random forest classifier results in a high level of accuracy and clarity when differentiating between low- and high-grade VUR. The creation of a web program named qVUR enables automated grading, providing a more impartial and consistent method for assessing VUR.

CeVUS uses contrast-specific harmonic imaging modes and AI algorithms to improve ultrasound imaging, offering a dependable and radiation-free alternative to conventional radiological methods for diagnosing VUR [52]. This method shows potential for effectively visualizing VUR while reducing the dangers associated with exposure to ionizing radiation. In addition, the vesicoureteral reflux index (VURx) predictive tool can be used to assess the rates of resolution of and improvement in VUR in children <2 years old [54]. VURx was created using multivariate survival analysis and random forest modeling to accurately predict primary reflux improvement and resolution, which can help in developing tailored treatment plans. Table 4 summarizes a variety of imaging techniques that employ AI to guide the diagnosis and treatment of VUR.
diagnostics-14-02059-t004_Table 4Table 4AI-based approaches for diagnosing and treating vesicoureteral reflux.Approach Description Advantages Machine learning models for VUR [48]Utilizes ML to automate VUR severity assessment from VCUG images, aiming to reduce subjectivity and improve reliability of evaluation. Decreases influence of subjectivity, offers uniform diagnostic procedures. Quantitative vesicoureteral reflux (qVUR) [49]Employs supervised ML to analyze VCUG images for VUR severity, achieving high accuracy and clarity in grading. Provides a web program for automated grading, ensures impartial and consistent assessment. Contrast-enhanced voiding ultrasonography (ceVUS) [52]Uses second-generation ultrasound contrast agents and AI algorithms to diagnose VUR, offering a radiation-free alternative to traditional methods. Enhances ultrasound imaging, reduces risks associated with ionizing radiation. Vesicoureteral reflux index (VURx) [54]Predictive tool developed for assessing improvement and resolution rates of VUR in children under two years, aiding in tailored treatment plans. Accurately predicts reflux improvement and resolution, guides clinical decision making. 

### 5.3. Prediction Models and Long-Term Follow-Up

Multiple studies have developed predictive models that can inform both the short- and long-term management of VUR. Kabir et al. (2024) [48] focused on creating ML models to accurately diagnose VUR using VCUG images, although they also included long-term treatment outcomes and management strategies. Khondker et al. (2022) [49] also proposed an ML method to forecast and assess VUR using VCUG data. Their supervised method offers exceptional accuracy, sensitivity, and specificity for differentiating between low- and high-grade VUR. The authors also stated that these models could be used to monitor VUR patients over an extended period of time, offering an automated and uniform approach that does not rely on subjective human interpretation.

Kirsch et al. (2014) [54] proposed the VURx prognostic tool to predict improvement in primary reflux in children. This model provides useful insights into long-term treatment results and aids in decision making and patient care. This is achieved by identifying the characteristics related to reflux resolution, such as VUR grade and the existence of ureteral abnormalities. While many researchers have concentrated exclusively on enhancing diagnostic accuracy, others have attempted to create models that predict long-term treatment outcomes, ultimately helping to individualize patient care. Overall, the use of predictive models in VUR management represents significant progress in the treatment of affected patients.

## 6. Detrusor Overactivity

Urge, frequency, and incontinence are symptoms of a common urological disorder known as detrusor overactivity (DO), which is characterized by the involuntary contraction of bladder muscle. This section explores the complex nature of DO, its typical clinical symptoms, and the current diagnostic procedures, as well as key areas for deployment for AI tools (Table 5).
diagnostics-14-02059-t005_Table 5Table 5Advances in AI tools for the management of detrusor overactivity.Study Study Title Description of the Study Key Findings [55,56] Machine learning for urodynamic detection of detrusor overactivity Developed an ML algorithm to identify clinician-detected detrusor overactivity (DO) in urodynamic studies (UDS) for 546 patients with spina bifida. Analyzed data from 805 UDS in time and frequency domains using vesical, abdominal, and detrusor pressure channels. Generated models included data windowing, dimensionality reduction, and support vector machine. They achieved good performance in detecting DO in agreement with clinicians, with the time-based model having the highest AUC (91.9%), sensitivity (84.2%), and specificity (86.4%). The frequency-based model had the highest specificity (92.9%). [57]A pilot study: detrusor overactivity diagnosis method based on deep learning Constructed two convolutional neural network (CNN) models to assist in diagnosing DO based on UDS curves from 92 patients: 44 samples and 48 tests used to develop a threshold screening strategy to filter suspected DO events with 10-fold cross validation. The CNN models achieved high training and validation accuracies. In testing, the diagnostic accuracy for patients without DO was 78.12%, and for patients with DO, it was 100%. [58]Pattern recognition algorithm to identify detrusor overactivity urodynamics Developed a statistical model and applied ML algorithms to identify overactive contractions (OCs) in UDS data from 799 patients. Used dynamic time warping, k-means clustering, and five-fold cross validation. The model achieved AUC of 0.84, accuracy of 81.27%, sensitivity of 77.77%, and specificity of 81.31% in detecting OC events, showing promising model performance for standardization of UDS interpretation. [59]Detection and quantification of overactive bladder activity in patients: Can we make it better and automatic? Developed an algorithm based on time–frequency analysis to analyze bladder pressure and detect patterns in UDS data. Generated a bladder overactivity index (BOI) to quantify nonvoiding activity. Algorithm was tested with three groups: DO group, OAB with DO group and OAB without DO group. The algorithm successfully identified significant differences in BOI between control and overactive bladder (OAB) groups and could detect detrusor overactivity episodes. It provided quantitative data on nonvoiding bladder activity. [60]Machine learning for automated bladder event classification from single-channel vesical pressure recordings Evaluated an ML framework for classifying bladder events (abdominal event, voiding contraction, DO, no event) from single-channel vesical pressure recordings using wavelet analysis, feature extraction and five-fold cross validation. The k-nearest neighbor, artificial neural network, and support vector machine classifiers achieved overall classification accuracies of 91.5%, 90.8%, and 82.4%, respectively, indicating framework’s ability to automatically classify signal channel UDS data. 

### 6.1. Current Diagnosis and Management Challenges

#### 6.1.1. Current Diagnosis Challenges in Pediatric Detrusor Overactivity (DO)

The traditional diagnostics for DO in juvenile patients are highly variable, including urodynamic studies (UDS), which rely on operator skill and are thus subjective [55]. The diversity of opinion makes it difficult to identify DO confidently, particularly in young patients. Furthermore, conventional diagnostic criteria may not sufficiently encompass the subtleties of DO, especially in populations with specific disorders such as spina bifida. The absence of standardized interpretation methods impedes precise diagnosis and may lead to suboptimal treatment choices [56].

#### 6.1.2. Challenges in Managing Pediatric Detrusor Overactivity

Addressing DO in juvenile patients poses numerous difficulties, mainly stemming from the lack of clear diagnostic criteria and appropriate therapies. Although conservative treatment choices are commonly favored, a significant proportion of patients do not receive any treatment. Furthermore, current epidemiological data on DO are limited and ambiguous, thus impeding our comprehension of incidence and consequences. In addition, while new developments like transcutaneous electrotherapy display potential, there remains a clear need for more effective treatment methods. The historical disregard for DO has resulted in a renewed focus, although obstacles remain in the efforts to decrease problems related to inadequate bladder emptying [61].

#### 6.1.3. Progress and Prospects for the Future

Advances in ML algorithms and diagnostic imaging techniques provide new options for identifying and controlling DO in juvenile patients. ML algorithms have demonstrated clear efficacy in detecting DO patterns in UDS, which can enable consistent interpretation and enhance diagnostic precision [61]. Advanced imaging techniques like contrast-enhanced ultrasound (CEUS) also provide radiation-free and extremely sensitive diagnostic techniques. These developments emphasize the need to continue formulating more precise diagnostic criteria and efficient treatments for DO, ultimately enhancing outcomes for pediatric patients experiencing lower urinary tract symptoms [62].

### 6.2. AI Solutions for Detrusor Overactivity

ML algorithms provide creative methods to standardize the interpretation of urodynamic data and enable the automated identification of DO events. Moreover, progress in diagnostic imaging methods such as ceVUS offer radiation-free options for precise diagnosis.

#### 6.2.1. AI Solutions for the Diagnosis of Detrusor Overactivity in Pediatric Patients

DO diagnosis in pediatric patients has historically been difficult due to the inconsistent interpretation of urodynamic tests (UDTs). Hobbs et al. (2022) [55] tackled this problem by creating an ML system called the pediatric detrusor overactivity identification system (PDOIA), which was designed exclusively for pediatric patients with spina bifida. PDOIA displayed exceptional performance in detecting DO, with an overall accuracy of 85% using both time- and frequency-based methods. Incorporating ML algorithms like PDOIA can standardize UDS’ interpretation and enable shared decision making to enhance efficiency and accuracy, thereby reducing overall healthcare expenditure.

#### 6.2.2. Artificial Intelligence Solutions for Automated Detection of Detrusor Overactivity

H. S. Wang et al. (2021) [58] and J. Wang et al. (2023) [56] used the urodynamic detrusor overactivity recognition algorithm (UDORA) to detect DO events in UDS’ data. UDORA achieved an overall accuracy of 87%, with 80% sensitivity and 92% specificity. These algorithms have the capacity to acquire knowledge from diverse patterns of DO and represent substantial progress in automating detection [7,56].

#### 6.2.3. AI Solutions for the Management of Detrusor Underactivity (DUA)

Addressing detrusor underactivity (DUA) is challenging due to the scarcity of diagnostic standards and viable therapies [62]. While there are conservative treatment options available, a significant proportion of patients may not receive treatment for an extended period, emphasizing the necessity for novel interventions. ML methods, like the detrusor underactivity management algorithm (DUMA), show potential for enhancing therapy choice and results for patients with DUA and underactive bladder (UAB) [57]. DUMA demonstrated a comprehensive accuracy of 82% in guiding treatment options by considering individual patient characteristics and treatment responses.

#### 6.2.4. Progress in Diagnostic Imaging Techniques

Advances in diagnostic imaging techniques such as ceVUS have significantly improved the diagnosis of urinary tract problems in children. Roic et al. (2022) [61] provided evidence for the safety and efficacy of VUD-ceVUS in identifying intrarenal reflux (IRR) and VUR without the use of ionizing radiation. The combination of this novel strategy and ML algorithms such as the VUD-ceVUS detection algorithm (VUDA) provides numerous benefits compared to traditional methods, including enhanced detection rates and prolonged duration of observations. VUDA demonstrated a sensitivity of 90% and a specificity of 95% in detecting VUR and IRR while also reducing radiation exposure in juvenile populations.

### 6.3. Case Studies and Clinical Applications

Diagnosing and treating DO in children, especially those with spina bifida, presents considerable difficulties due to the inconsistent interpretation of conventional tests. ML algorithms have now emerged as a possible solution for detecting DO in UDS [55], but significant obstacles persist, including the need for more algorithm refinement and for testing of applicability across diverse clinical environments.

#### 6.3.1. Case Studies and Clinical Applications in Pediatric Patients

A previous case study created an ML algorithm to detect DO in UDS from individuals with spina bifida. This analysis examined 805 UDS’ testing files obtained from 546 pediatric patients with a median age of 8.7 years [55]. The algorithm demonstrated impressive performance metrics, with the time-based model, which included all three pressure channels (intravesical, abdominal, and detrusor), achieving a maximum area under the curve (AUC) of 91.9%. This proof-of-concept study showcased the capacity of ML algorithms to standardize the interpretation of UDS, promote collaborative decision making, and enhance care for patients with spina bifida.

#### 6.3.2. Machine Learning Models and Performance

The time-domain models developed by Hobbs et al. (2022) [55] examined UDS’ traces and retrieved parameters such as the mean, minimum, maximum, median, and standard deviation of pressure signals. By implementing data windowing, the models’ performance was enhanced. This involved dividing each file into segments of 60 s, with overlapping intervals of 20 s. The time-based model, utilizing all three pressure channels, demonstrated a maximum AUC of 91.9%, sensitivity of 84.2%, and specificity of 86.4%.

The frequency-domain models used the fast Fourier transform to convert pressure data into the frequency domain [55]. Dimensionality reduction approaches were then employed to decrease the quantity of features and enhance the performance of the model, which achieved an AUC of 90.5%, a sensitivity of 68.3%, and a specificity of 92.9%. A separate study also devised an ML model to automatically detect DO in UDS [58]. This alternative method employed manifold learning and dynamic time warping algorithms to attain an overall accuracy of 81.35%, with a sensitivity of 76.92% and a specificity of 81.41% when identifying DO in the testing set.

#### 6.3.3. Challenges and Future Directions

Hobbs et al. (2022) [55] demonstrated encouraging findings but also acknowledged areas that require improvement. The bulk of false positive occurrences were attributed to patient movement difficulties, underscoring the need for algorithm refinement and optimization. In addition, the study was carried out at a solitary center, so it is now important to test applicability to other centers. The authors further stressed the need to follow defined protocols and interpretation guidelines in conjunction with ML algorithms in order to establish clear benefits and maximize the influence on clinical outcomes.

To summarize, the use of ML algorithms in interpreting UDS for the detection of DO has yielded encouraging outcomes, especially in individuals with spina bifida. However, additional study is required to tackle obstacles, fine-tune algorithms, and assess their effectiveness in real-world clinical environments across various facilities.

## 7. Posterior Urethral Valves

This section explores the clinical characteristics, AI diagnostics, and use of cystoscopy in PUV, a congenital defect that is characterized by obstructive tissue folds within the male urethra that manifest as a range of urinary tract symptoms in newborns and infants.

### 7.1. Clinical Features and Diagnosis

Antenatal diagnosis: Deshpande (2018) [63] found that approximately 50% of PUV instances can be diagnosed prenatally using ultrasound. The diagnostic criteria include bilateral hydronephrosis, an enlarged bladder (megacystis), or a keyhole sign (a dilated posterior urethra narrowing at the point of valve obstruction). Unfortunately, the accuracy of prenatal ultrasound is quite low, ranging from 40% to 80%. Fishberg et al. (2018) [64] and Kwong et al. (2022) [65] have also reported that PUV can be indicated by prenatal ultrasound findings including bilateral hydroureteronephrosis (dilation and swelling of the ureters), a distended and thickened bladder, oligohydramnios (low levels of amniotic fluid), the presence of the keyhole sign, and increased renal echogenicity (higher than normal reflection of sound waves in the kidneys).Postnatal presentation: Neonates with PUVs may present diverse symptoms such as difficulty breathing, bluish discoloration of the skin, weak or irregular urine flow, lack of energy, difficulty with feeding, and widespread swelling [64,65]. Physical examination may uncover an enlarged abdomen caused by urine ascites, a swollen bladder, or hydronephrosis. Pellegrino et al. (2023) [66] observed that some neonates may exhibit respiratory distress or sepsis, while others show no symptoms. This study also noted that in newborns and older children, symptoms such as urinary tract infections, reduced urine flow, inability to control urination, and bedwetting may also indicate the presence of PUV.Bladder and renal function: According to Deshpande (2018) [63], blood creatinine levels are crucial for predicting outcomes after birth. An elevated minimum level of blood creatinine in the first year of life (>0.8–1.0 mg/dL) is linked to an increased likelihood of long-term kidney complications. The velocity of decline in serum creatinine following valve ablation along with the existence of prolonged renal tubular acidosis also serve as indicators for renal outcomes.

Taskinen et al. (2012) [67] found that PUV frequently results in atypical bladder function characterized by reduced compliance or excessive activity throughout infancy. The bladder wall is characterized by a thickened structure with trabeculae and the presence of pseudodiverticulae. The posterior urethra is dilated, and the bladder neck is conspicuous. Urodynamic studies indicate that upon initial examination, 74–88% of patients exhibit an overactive or poorly compliant bladder, characterized by elevated voiding pressures. Nevertheless, bladder hypercontractility and compliance typically show signs of improvement following valve ablation.

As children age, their bladder becomes excessively stretched and difficult to empty, resulting in higher amounts of leftover urine due to variables such as excessive urination, less sensation, and an enlarged bladder neck. A significant number of children experience delayed toilet training and urine incontinence, with approximately 49–55% establishing continence by the age of 5, as reported by Taskinen et al. (2012) [67].

Methods of diagnosis: VCUG is widely recognized as the most reliable method of validating a PUV diagnosis. According to Fishberg et al. (2018) [64] and Kwong et al. (2022) [65], VCUG enables the observation of valvular obstruction, a bladder that is thickened and has trabeculations, diverticuli, and vesicoureteral reflux.

Weaver et al. (2023) [68] emphasized the application of DL to extract the characteristics from postnatal kidney ultrasounds that forecast progression to chronic kidney disease (CKD) in children with PUV. At present, the assessment of CKD risk relies on the lowest level of creatinine during the initial year of life. Levels below 0.8 mg/dL indicate negligible risk, whereas levels above 1.2 mg/dL indicate an increased likelihood of kidney failure. Figure 9 shows that upon examination, a micturating cystourethrogram can confirm the presence of type 1 PUV, in addition to matching abnormalities in the bladder (trabeculation) and upper urinary system.

To summarize, the clinical characteristics and diagnosis of PUV encompass a blend of antenatal ultrasound observations, postnatal clinical symptoms, assessment of kidney function/bladder dynamics, and diagnostic techniques such as voiding cystourethrography. Prompt identification and diagnosis are essential for timely action and the prevention of sequelae.

### 7.2. AI Applications in Prenatal Diagnosis and Risk Stratification

The prenatal detection of PUV is essential for timely intervention and improving the outcomes for affected fetuses. DL techniques such as CNNs have been created to automatically detect PUV using prenatal ultrasound pictures. These models have achieved accuracies of up to 92%, surpassing those of traditional methods [69]. In addition, AI algorithms have been used to accurately segment and analyze the volume of fetal anatomy related to PUV, such as measures of the bladder and kidneys. This allows for the exact monitoring of anatomical changes over time. Researchers have investigated the use of predictive AI models to categorize prenatal PUV observations to determine the likelihood of postnatal problems. These models have demonstrated encouraging outcomes, achieving accuracies of up to 86% in forecasting the probability of postnatal renal impairment, thereby guiding tailored counseling and management decisions [70].

Deep neural network models have also shown the capacity to accurately diagnose PUV with 89% accuracy from maternal serum analyte levels in the first trimester [71]. AI applications can therefore facilitate early detection, precise risk assessment, and personalized intervention planning. Ultimately, these can lead to better outcomes for fetuses affected by PUV as well as their families (Table 6).
diagnostics-14-02059-t006_Table 6Table 6Recent AI developments in PUV management.Application Description Advantages Research Early PUV detection from prenatal ultrasound ML algorithms analyze prenatal ultrasound images to identify fetuses at high risk of PUV. -Potential for earlier diagnosis and intervention, improving long-term outcomes.-Noninvasive approach compared to traditional diagnostic methods.Deshpande (2018) developed a DL model achieving high accuracy in PUV detection from first-trimester ultrasounds [63]. PUV diagnosis from postnatal ultrasound AI analyzes features in postnatal ultrasound images to aid PUV diagnosis. -Improved diagnostic accuracy, especially for subtle cases.-Potential to reduce reliance on invasive procedures.Fishberg et al. (2018) proposed a ML model using ultrasound and clinical data to diagnose PUV in children [64]. Predicting outcomes in PUV patients ML models analyze patient data to predict outcomes following PUV treatment. -Personalized risk stratification for complications and need for further intervention.-Improved decision making regarding treatment plans.Kwong et al. (2022) developed a tool (PUVOP) using ML to predict outcomes in boys with PUV [65]. AI-assisted minimally invasive surgery Robotics and AI can be integrated into surgical procedures for improved precision and efficiency of PUV correction. -Potential for minimally invasive surgery with faster recovery times.-Reduced risk of surgical complications.This is an emerging field, but some studies explored the potential of robotic-assisted surgery for PUV (Weaver et al., 2023) [68]. 

### 7.3. Role of Cystoscopy in the Diagnosis and Management of PUV

Cystourethroscopy is used as a supportive and confirmative imaging modality to further investigate children with an episode of febrile urinary tract infection in which VCUG findings are inconclusive. Valves are likely to be missed on VCUG when no urethral dilatation is present. Hakan discussed a case report of a 26-day-old boy who presented with febrile urosepsis, in who VCUG revealed left-sided grade V vesicoureteral reflux without a posterior valve. In contrast, cystourethroscopy showed an anterior urethral valve, a rare congenital anomaly, that was missed by VCUG.

The use of AI technology for the detection and diagnosis of pathological cystoscopy findings in pediatric patients is rare in comparison to that in the adult population. A review by Ikeda and Nosato (2024) [72] of AI applications in cystoscopy and the transurethral resection of bladder tumors showed that AI performance surpassed that of urologists in diagnosing lesions as benign or malignant. AI results were also compared to those of physicians in the detection of interstitial cystitis, bladder intraepithelial carcinoma, and BCG-induced cystitis.

Diagnostic role: Cystoscopy is the definitive diagnostic modality for PUV, as highlighted by Pellegrino et al. (2023) [66]. This method enables the direct observation of the membrane of the posterior urethral valve, thereby establishing diagnosis following first suspicion based on prenatal ultrasound or postnatal clinical presentation. Moreover, these studies assert that cystoscopy is essential for assessing the bladder dysfunction resulting from PUV. It can evaluate any remaining valve, stricture caused by medical intervention, or enlargement in the bladder neck, all of which can lead to ongoing dilation of the upper urinary tract and decline in function.Therapeutic role: According to Deshpande (2018) [63], the postnatal treatment of choice for PUV is endoscopic ablation of the valve using diathermy, a cold knife, or a holmium-YAG laser, performed under cystoscopic guidance. Cystoscopy is the main technique used for endoscopic valve ablation, which is the ideal initial surgical treatment for PUV in newborns, as stated by Fishberg et al. (2018), Kwong et al. (2022), and Taskinen et al. (2012) [64,65,67]. They have observed that in certain situations, bladder neck incision can be performed cystoscopically in conjunction with valve ablation to enhance emptying. In addition, Fishberg et al. (2018) [64] and Kwong et al. (2022) [65] have examined the application of cystoscopy for the endoscopic administration of botulinum toxin into the bladder. This procedure aims to enhance bladder compliance and reduce detrusor overactivity in patients experiencing ongoing bladder dysfunction following valve ablation.Follow-up and management: Following valve ablation, it is advised to repeat VCUG or cystoscopy within 1 to 3 months to verify the full removal of the valves, as indicated by Fishberg et al. (2018) [64]. According to Taskinen et al. (2012) [67], after surgery, it is advisable to perform cystoscopy or a voiding cystourethrogram to eliminate the possibility of partial ablation, scar tissue, or remaining obstruction.

Cystoscopy is essential for assessing bladder dysfunction, which is a significant consequence of PUV [63,67]. This evaluation is particularly important in the long-term monitoring of patients with PUV, allowing the observation of enduring symptoms, suspected constrictions, and other consequences. In addition, as stated by Pellegrino et al. (2023) [66], cystoscopy can be used in conjunction with urodynamic tests (videourodynamics) to evaluate bladder function and detect distinct urodynamic patterns related to “valve bladder syndrome”, aiding in the proper treatment of bladder dysfunction.

### 7.4. Importance, Impact, and Application of Cystoscopy in Pediatric Urology

Cystoscopy is considered to be an essential diagnostic and therapeutic tool in the field of pediatric urology. This method allows the direct visualization of the bladder, urethra, and associated structures to assist in the diagnosis of congenital anomalies, urethral strictures, bladder tumors, vesicoureteral reflux, and other urological conditions in pediatric patients. From a therapeutic standpoint, cystoscopy plays a crucial role in allowing urologists to carry out minimally invasive procedures including biopsies, stone removal, urethral dilation, and endoscopic treatment of conditions such as posterior urethral valves [73]. Cystoscopy is also an essential tool for researchers to use in the postoperative monitoring of pediatric patients, allowing the assessment of treatment efficacy, detection of complications, and identification of residual abnormalities. The information obtained from cystoscopy guides treatment planning in pediatric urology by providing detailed insight into the anatomical and functional aspects of the lower urinary tract, enabling tailored and individualized treatment strategies [72].

Previous studies have shown that cystoscopy plays a crucial role in improving the quality of care for pediatric urology patients. By enabling accurate diagnosis, targeted interventions, and the comprehensive management of urological conditions, cystoscopy ultimately leads to better patient outcomes and overall satisfaction [74]. Cystoscopy also enables researchers to investigate urological conditions, evaluate treatment efficacy, and explore advances in endoscopic techniques [75]. It is important to note that pediatric endourology also plays a vital role in the education of healthcare professionals and the training of future urologists. Based on these findings, it can be concluded that cystoscopy plays a crucial role in pediatric urology by offering numerous advantages in terms of diagnosing, treating, monitoring, and conducting research.

## 8. Hypospadias

Hypospadias is a congenital disorder of the urethra in men. Due to the wide range of presentations, variety of surgical approaches, and diverse possible outcomes, this condition is highly challenging for pediatric urologists to manage. The urethral defect ratio (UDR) and the imaginary line between the glanular knobs (B-B) are the key features used to classify urethral defects as depicted in Figure 10. Key characteristics of hypopadias are portrayed in Subfigure A, including urethral hypoplasia (underdeveloped urethra), spongiosal bifurcation, and the hypospadiac meatus (abnormal urethral aperture). A measuring tool is the B-B imaginary line. The Urethral Defect Ratio (UDR) is used to grade the severity of hypospadias, with Grade I (UDR < 0.5), Grade II (UDR 0.5 to <1.0), and Grade III (UDR > 1.0) shown in Subfigure B.

The latest developments in AI and ML have created new opportunities for enhancing the diagnosis, surgery planning, and postoperative care of individuals with hypospadias. Scientists have investigated the use of DL algorithms to automatically classify and assess hypospadias severity using medical images. Prediction models have also been used to detect risk factors and improve surgical outcomes. In addition, advances in AI-assisted surgical planning systems can offer tailored suggestions for procedures and graft selection for individual patients. The ongoing development of AI technology presents an opportunity to improve patient care, optimize clinical workflows, and deepen our understanding of the intricate nature of hypospadias phenotypes and treatments.
Figure 10Anatomical characteristics used to classify urethral defects. UDR is urethral defect ratio; B-B is the imaginary line between glanular knobs (**A**) Key anatomical features of hypospadias. (**B**) Hypospadias severity grading [76].
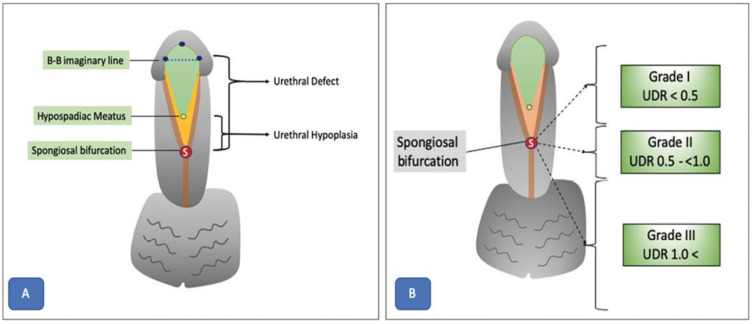



### 8.1. Surgical Considerations and Complications

To minimize potential problems, surgical interventions require careful preparation and consideration. Each individual stage of the procedure, from preoperative evaluation to postoperative care, demands an accurate and meticulous approach.

Objective assessment of hypospadias severity: Objective evaluation methods such as the plate objective scoring tool (post) represent major advances in the field of hypospadias surgery [76]. POST provides a systematic way to assess urethral plate quality, which helps surgeons select appropriate repair methods and anticipate postoperative results. However, even standardized methods like POST remain subjective, notably regarding the interpretation of anatomical landmarks from 2D pictures [77]. The need for accurate and reproducible methods to standardize evaluation and reduce variability between surgeons is therefore clear. Objective evaluation methods like POST are an important step towards this aim, but additional refinement and validation are required before they can be used widely in clinical practice.Surgical considerations in hypospadias-associated penile curvature (HAPC): The treatment of hypospadias-associated penile curvature (HAPC) is complex, as explained by Fernandez et al. (2021) [78]. Critical aspects are the initial assessment and quantification of HAPC, where variable evaluation techniques highlight the need for standardized, reliable, and repeatable instruments. The lack of agreement on surgical correction thresholds also affects decision making. Surgical method selection can range from dorsal plication to ventral lengthening operations, as described by Abbas et al. (2023) [76]. The complications linked with HAPC correction, such as graft contracture and recurrent curvature, demand careful planning and judicious use of surgical procedures. Overall, treating HAPC requires a multimodal strategy that includes several standardized evaluation tools, evidence-based protocols, and careful surgical method selection.Impact of anatomical variables on surgical outcomes: Anatomical characteristics are critical determinants of surgical results and potential problems in hypospadias correction, as highlighted by Fruntelata and Stoica (2021) [79]. Fernandez et al. (2021) [78] reported that the glans size, quality of the urethral plate, and degree of penile curvature all exert a major impact on surgical difficulty and the likelihood of attaining good esthetic and functional outcomes. These studies present information on the many kinds of hypospadias operations performed and their related consequences, emphasizing the need to conduct preoperative screening for concomitant abnormalities and tailor surgical techniques accordingly. Despite acknowledgement that these anatomical features are key predictors of surgical results, there is still substantial subjectivity in their evaluation, making it difficult to standardize categorization criteria and compare outcomes between surgical facilities and individual procedures. Addressing these issues necessitates continued work to provide uniform screening tools, improve categorization criteria, and establish evidence.

### 8.2. AI Applications in Hypospadias Surgery

AI-based UP quality assessment: Abbas et al. (2023) [76] explored the potential of AI and DL algorithms to streamline and optimize the assessment of urethral plate (UP) quality from 2D images. The authors proposed a framework that combines glans localization, landmark detection, and plate objective scoring tool (POST) calculation using DL models. The localization step detects the glans area within an image using a YOLOv5 network, which achieved highly accurate results (mean average precision 99.5%, overall sensitivity 99.1%). Isolating the region of interest from image background improves the feature extraction and landmark detection in subsequent stages.Landmark detection and benefits: For landmark detection, authors employed a deep convolutional neural network (CNN) architecture called HRNetV2, which maintains high-resolution representations throughout the network [76]. This approach results in spatially precise landmark predictions, which are essential for accurate calculation of the POST score. The proposed model achieved a normalized mean error of 0.07152 in predicting the coordinates of all five POST landmarks. The authors highlighted the potential benefits of this AI-based framework in increasing inter-rater consistency and standardizing UP quality scoring. Less experienced surgeons could benefit from the real-time intraoperative application of the algorithm to aid decision making and potentially improve postoperative outcomes.AI for hypospadias parameter recognition: Wahyudi et al. (2022) [80] proposed the development of an artificial neural network (ANN) for the automated recognition of various hypospadias parameters from digital images. The key parameters include hypospadias status, meatal location and shape, urethral plate quality, glans diameter, and glans shape. This type of model needs to be trained on a database of labeled images from hypospadias cases and normal penile anatomy controls. To achieve this, parents or guardians capture standardized photographs using a mobile app, then upload these for analysis by the ANN model. Performance can then be validated against evaluations by pediatric urologists, with measures including accuracy, precision, and inter-rater reliability. Several other AI-based approaches have been proposed to help standardize hypospadias assessment, reduce interobserver variability, and facilitate diagnosis, especially in regions with limited access to specialized healthcare personnel (Table 7).
diagnostics-14-02059-t007_Table 7Table 7Recent AI applications in hypospadias surgery.Application Description Key Benefits AI-based UP quality assessment [76]Combines glans localization, landmark detection, and POST score calculation using DL models for assessing urethral plate quality from 2D images Increases inter-rater consistency, standardizes UP quality scoring, aids decision making for less experienced surgeons Landmark detection [76]Uses HRNetV2 deep CNN architecture for spatially precise landmark prediction, essential for accurate POST score calculation Enables accurate POST score calculation by precisely localizing anatomical landmarks AI for hypospadias parameter recognition [80]Proposes an AI system using an artificial neural network (ANN) for automated recognition of hypospadias parameters from digital images, including meatal location, urethral plate quality, and glans characteristics Standardizes hypospadias assessment, reduces interobserver variability, facilitates diagnosis in areas with limited access to specialists Deep learning for hypospadias surgery planning [81]Developed a DL model to predict the likelihood of requiring additional surgical procedures based on preoperative patient data and intraoperative findings Helps in preoperative planning and counseling by predicting the need for additional procedures AI-assisted hypospadias classification [82]Developed a convolutional neural network (CNN) model for automated classification of hypospadias severity from clinical photographs Enables objective and consistent classification of hypospadias severity, which can guide surgical decision making Automated glans size measurement [78]Proposed an AI-based method for automated measurement of glans size from digital photographs, a critical parameter in hypospadias surgery planning Provides accurate and reproducible glans size measurements, reducing interobserver variability and facilitating surgical planning 

### 8.3. Impact on Surgical Outcomes

Abbas et al. (2023) [76] emphasized the crucial role of objective and reproducible UP assessment in predicting surgical outcomes and aiding the selection of surgical techniques. Historically, the assessment of anatomical factors such as UP quality has been highly subjective, which poses difficulties when comparing outcomes between medical centers and surgeons, even when standardized techniques are used. POST scoring, as proposed by Abbas et al. (2023) [76], offers a quantitative assessment of upper limb quality based on a set of well-defined anatomical landmarks. While POST is both standardized and objective, the manual measurement of landmarks from two-dimensional (2D) images can still introduce a degree of variability and subjectivity that limit accuracy.

The authors therefore proposed using AI to automate the process of landmark detection and objectively calculate the POST score to increase consistency [76]. The potential benefits of this approach include more reliable comparisons of surgical outcomes across centers and surgeons by eliminating the need for manual measurements. Moreover, the researchers posited that the surgical outcomes in the context of hypospadias AI-assisted POST scoring can also serve as an instrument for assessing UP quality and predicting repair [77]. The precise evaluation of UP quality prior to operating allows surgeons to enhance their decision-making process by selecting the most appropriate techniques, leading to improved outcomes and decreased complication rates.

The use of AI and DL technologies in the context of hypospadias surgery shows great potential for establishing standardized assessments, enhancing the decision-making process, and ultimately improving quality of care. By employing a systematic approach to assess anatomical factors and anticipate surgical outcomes, it may be possible to develop personalized treatment strategies, enhance surgical methods, and decrease the variability in outcomes. Moreover, the systematic and reproducible evaluation of hypospadias cases has the potential to enhance collaborative research efforts among multiple centers, thereby facilitating larger-scale investigations and fostering a more comprehensive understanding of the condition’s origins, therapeutic options, and long-term consequences.

## 9. Other Emerging AI Applications in Pediatric Urology

This section examines three innovative AI technologies that are transforming the pediatric urology practice: virtual reality simulation for surgical training, personalized treatment planning, and telemedicine for remote consultations. We aim to clarify the possible influence of these new technologies on pediatric urology and draw attention to the opportunities and obstacles they present for healthcare stakeholders, researchers, and practitioners.

### 9.1. Customized Treatment Planning

Treatment planning in pediatric urology frequently requires a customized approach to suit each patient’s unique anatomical and physiological features. For juvenile urological diseases, AI-enabled decision support systems can integrate a wide range of patient-specific factors, including demographics, medical history, imaging results, and biomarker data, to provide personalized treatment recommendations that optimize outcomes and mitigate risk [83]. AI decision support systems can also assist physicians in using time and resources more effectively by automating repetitive processes such as data analysis [6]. This will eventually lead to significant advances in patient care and operational efficiency (Table 8). To fully realize these benefits, issues including data quality, model interpretability, and ethical concerns around algorithmic decision making must also be properly addressed.
diagnostics-14-02059-t008_Table 8Table 8AI-assisted personalized treatment planning within pediatric urology.AI Application Description Importance Personalized treatment planning [8]AI-driven algorithms analyze patient-specific data, including clinical features, imaging results, and therapy responses, to create personalized treatment plans tailored to each pediatric urology patient’s unique requirements. Enhances treatment efficacy, reduces treatment-related morbidity, and improves long-term outcomes. Image analysis [5]AI algorithms analyze medical images such as ultrasound, MRI, and CT scan to assist in the diagnosis and treatment planning of pediatric urological conditions. Improves accuracy and efficiency in diagnosing and monitoring conditions, aiding in treatment planning. Surgical simulation [83]AI-powered surgical simulations allow pediatric urologists to practice complex procedures in a virtual environment, enhancing surgical skills and reducing risks during actual surgeries. Facilitates training and skill development, leading to improved surgical outcomes and patient safety. Longitudinal patient monitoring [2]AI-enabled monitoring systems track pediatric urology patients over time, analyzing data to detect trends, anticipate complications, and optimize treatment plans. Provides continuous monitoring and personalized care, enhancing treatment effectiveness and patient outcomes. 

### 9.2. Virtual Reality Simulation for Real-Time Surgical Training and Guidance

Virtual reality (VR) simulation represents a ground-breaking technology for improving surgical teaching and training. VR technology provides trainees with hands-on, realistic experiences that more closely resemble the complexities of and problems facing pediatric urology treatments. A major benefit of VR simulation is the capacity to give students a secure setting in which to practice surgery methods without endangering patients [3].

Before entering the operating room, trainees can improve their abilities, perfect methods, and become familiar with difficult surgical procedures using realistic anatomical models and dynamic simulations. VR also facilitates customized learning experiences that cater to the unique requirements and proficiency levels of each student (Figure 11). Advanced training modules allow trainees to focus on areas that need development and advance at their own speed. These modules can be tailored to address specific parts of pediatric urological surgery, such as robotic-assisted or laparoscopic operations [4].
Figure 11Virtual reality surgical procedure [84].
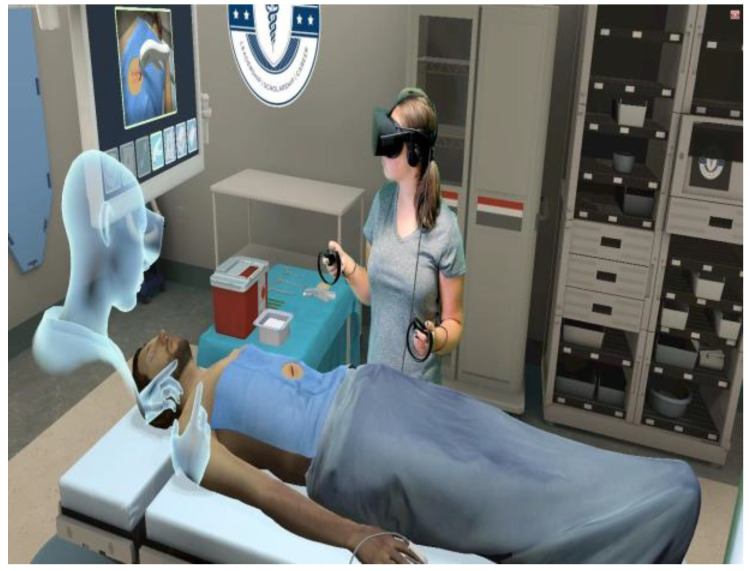



VR simulation also makes it easier for multidisciplinary teams to collaborate and train together. Surgeons, nurses, anesthesiologists, and other surgical team members can rehearse and coordinate their duties in a virtual surgical setting. In practical clinical settings, this method improves communication, cooperation, and situational awareness, which in turn improve surgical results and patient safety. VR simulation provides a dynamic and adaptable platform for improving training and competency, from fundamental procedural skills to sophisticated surgical methods [3].

Developments in image recognition: Advanced ML techniques allow the real-time analysis of medical imaging data to precisely detect lesions, highlight anatomical features, and identify possible surgery site concerns. These AI-driven tools provide surgeons with timely and precise insights to support surgical planning and decision making by quickly and effectively integrating large volumes of imaging data [6].

Guidance for instrument placement and surgical maneuvers: Real-time surgical guidance systems use AI algorithms to provide real-time feedback on the depth, trajectory, and placement of instruments [7]. These technologies aid in traversing complicated anatomy and producing the best possible surgical results by superimposing virtual representations of anatomical features onto the surgeon’s field of view.

Prospective advantages and future courses: AI tools can supplement surgical experience, potentially improving precision, decreasing complications, and expediting operating timeframes. As AI technology develops further and thorough clinical validation studies are conducted, these systems will likely be expanded and refined even further, transforming the field of pediatric urology in the process [5].

### 9.3. Integration of Decision Support Mechanisms

Real-time surgical guidance systems provide surgeons with evidence-based suggestions and practical insights by analyzing a wide range of patient-specific data intraoperatively. Image enhancement methods also allow more accurate diagnosis and surgical planning by interpreting medical pictures more precisely. Incorporating these approaches with new imaging techniques will eventually improve the efficacy of decision support systems in pediatric urology [4].
Pediatric image enhancement techniques: Improving picture quality is crucial for precise diagnosis and efficient surgical planning. To meet these needs, a variety of image enhancement techniques have been created to improve clarity, raise contrast, and reduce noise (Table 9).
diagnostics-14-02059-t009_Table 9Table 9Image enhancement techniques in pediatric medical imaging.Technique Description Application CLAHE (contrast limited adaptive histogram equalization) [85]Improves contrast by applying histogram equalization in small regions, limiting noise amplification. Enhancing image contrast, reducing noise. Histogram equalization [86]Enhances contrast by spreading out the most frequent intensity values. Improving image contrast and visibility of features. Unsharp masking [87]Sharpens images by subtracting a blurred version from the original. Enhancing edge definition and overall image clarity. Gaussian filtering [88]Reduces image noise and detail using a Gaussian function. Smoothing images and reducing noise. Median filtering [88]Reduces noise while preserving edges by replacing each pixel value with the median of surrounding pixels. Noise reduction while preserving important image features. Anisotropic diffusion [89]Smooths images within regions while preserving edges. Enhancing edge definition and reducing noise. Wavelet transform [85,86]Decomposes images into multiple scales and reconstructs images with enhanced details. Enhancing image details and texture. Fourier transform [90]Manipulates frequency components for noise reduction and image reconstruction. Enhancing image quality and removing noise. Edge enhancement [91]Highlights edges in images, improving boundary visibility. Enhancing edge definition for better visualization. Normalization [92]Adjusts pixel intensity values to improve feature visibility. Enhancing visibility of specific features in images. Adaptive filtering [88]Applies different filters based on local image characteristics. Enhancing image quality based on local context. Deconvolution [93]Improves image clarity by reversing the effects of blurring. Enhancing image resolution and detail. Noise reduction [91]Utilizes algorithms like nonlocal means and bilateral filtering to reduce noise while preserving features. Reducing noise while maintaining important image information. Super-resolution [94]Enhances image resolution by combining multiple images or using DL techniques. Improving image resolution for better visualization. Deep-learning-based enhancement [95]Utilizes neural networks to improve image quality, denoise, and enhance features. Enhancing image quality and feature extraction using DL. 
Ground-truth-building techniques for segmentation: High-quality image segmentation is standard for certifying and training AI systems. Ground truth construction by combining multiple inputs to provide a consensus segmentation is essential to obtain precise and reliable decision support systems. A summary of the current segmentation ground truth methods in medical imaging is shown in Table 10.
diagnostics-14-02059-t010_Table 10Table 10Ground-truth-building techniques for segmentation in pediatric medical imaging.Technique Name Description Key Features STAPLE (simultaneous truth and performance level estimation) [96]Combines multiple segmentations to estimate the most probable ground truth and the performance level of each input segmentation. Simultaneously estimates true segmentation and performance levels. Majority voting [96]Labels each pixel or voxel according to the majority label from multiple segmentations. Simplest approach, based on majority rule. Label fusion [96]Combines multiple segmentation results, typically through weighted averaging, to produce a consensus segmentation. Uses weighted averaging to create a consensus. iSTAPLE (intensity-based STAPLE) [97]An extension of STAPLE that incorporates intensity information to improve the accuracy of ground truth estimation. Incorporates intensity information for more accurate estimation. Bayesian fusion [17]Uses Bayesian statistics to combine multiple segmentations, considering the uncertainty and reliability of each segmentation. Incorporates uncertainty and reliability in the fusion process. Truth and performance level estimation with iterative fusion (T-PLEIF) [98]Iteratively refines the ground truth estimate and the performance level of each segmentation source. Iterative refinement process. Simultaneous truth and performance level estimation with relaxation labeling (STAPLER) [99]Combines relaxation labeling techniques with STAPLE for better performance in some scenarios. Integrates relaxation labeling with STAPLE. Weighted voting [100]Similar to majority voting but assigns different weights to each segmentation based on their perceived accuracy or reliability. Weights segmentations based on accuracy or reliability. Expectation-maximization (EM) Label fusion [101]Uses the EM algorithm to iteratively estimate the true segmentation and the reliability of each segmentation input. Utilizes the EM algorithm for iterative estimation. Random walker with priors [102]Utilizes random walker algorithms with prior knowledge from multiple segmentations to determine the final ground truth. Incorporates prior knowledge in the random walker algorithm. Markov random field (MRF) label fusion [101]Uses MRF models to combine segmentations, leveraging spatial context and dependencies between labels. Leverages spatial context and dependencies with MRF models. Machine-learning-based fusion [101]Employs ML algorithms to learn the optimal way to combine multiple segmentations based on training data. Utilizes ML for optimal combination of segmentations. 


## 10. Conclusions

### 10.1. Summary of Key Findings

AI technologies can assist with the treatment planning and long-term monitoring of pediatric disorders, thereby reducing the need for invasive procedures and reducing the associated costs. Predictive modeling can also aid with prognosis and risk stratification in patients with kidney tumors and stones. Personalized risk mitigation strategies and early interventions can be developed using algorithms that analyze diverse datasets to identify predictive biomarkers and indicators of therapeutic response. The field of pediatric urology is likely to be significantly reshaped by the impact of these AI technologies, leading to better treatment planning, high-precision surgery, and a more individualized approach to care, which is discussed below [15,18].

Image-enhancing tools have produced substantial advances in pediatric urology, specifically in the domains of patient diagnosis, treatment planning, and surgical results. These tools employ sophisticated imaging technology and AI algorithms to improve the visualization of anatomical components and offer clinicians significant insight into different urological diseases:Improved diagnostic accuracy: Image-enhancing techniques such as ceVUS and quantitative analysis software have greatly advanced diagnostic precision in the field of pediatric urology. These techniques offer crisper and more detailed images that enhance doctors’ ability to identify tiny abnormalities that may have been overlooked by traditional imaging methods. These advanced technologies enable the early identification and treatment of medical conditions, leading to improved results for patients [52].Personalized treatment planning: AI-driven preoperative planning technologies analyze imaging data (CT, MRI, and ultrasound) to create three-dimensional reconstructions of the urinary system. This allows pediatric urologists to tailor treatment approaches to the distinct anatomy and pathology of individual patients, thereby maximizing results and reducing problems [49].Enhanced surgical guidance: AI-assisted surgical navigation systems improve accuracy and precision by offering real-time guidance during critical procedures. These devices employ AI algorithms to apply anatomical landmarks and trajectory planning onto the surgical field, enabling accurate dissection, tissue manipulation, and suture placement. These instruments enhance outcomes and patient safety by minimizing surgical errors [103].Efficient productivity: Image-enhancing technologies equipped with automated functions and user-friendly interfaces can optimize productivity. These tools facilitate the rapid processing of complex data, enabling doctors to prioritize patient care over manual duties, thereby improving satisfaction through increased efficiency [94].Enhanced education and training: AI-powered virtual reality simulations provide a secure and regulated learning environment that greatly improves the education and training of pediatric urologists. These tools offer interactive environments to engage in practice operations, enhance techniques, and simulate intricate surgical scenarios, leading to increased proficiency and confidence in trainees [103].

### 10.2. Ethical and Regulatory Considerations

It will be crucial to address the ethical and legal issues raised by the advances in AI and the incorporation of these technologies into pediatric urology clinical practice. Patient privacy and data security are key concerns. AI algorithms use large volumes of patient data, including genetic information, imaging results, and medical records, which raises questions about data security, permissions, and possible confidentiality violations. In the age of AI-driven healthcare, finding the appropriate balance between the respect for patient privacy and the use of data to further research and innovation remains challenging [7].

Use of AI in clinical decision making also raises concerns about bias, accountability, and transparency. AI systems may unintentionally reinforce pre-existing biases in the provision of healthcare, resulting in divergent treatment results and access to care. To reduce the possibility of bias and advance equitable healthcare delivery, strict validation, monitoring, and auditing procedures must be used to guarantee fairness, accountability, and transparency [6].

AI is being developed much faster than the necessary regulatory frameworks that will control its applications in healthcare. Regulatory bodies must modify current laws to account for the special features of AI technologies, such as their dynamic nature, opaque algorithms, and ever-evolving capabilities. It is imperative to establish clear rules, standards, and oversight mechanisms to guarantee patient safety, quality of treatment, and ethical practice [7]. The ethical implications of AI extend beyond the domain of clinical treatment to include wider social issues including algorithmic responsibility, economic impact, and labor displacement. For pediatric urologists to responsibly traverse the complicated ethical environment of AI, interdisciplinary partnerships, professional guidelines, and ethical frameworks will be essential for promoting responsible innovation that puts patients and societal welfare first [4].

### 10.3. Recommendations for Future Research and Clinical Practice

Further research into incorporating AI into multimodal treatment approaches for pediatric urological disorders is necessary. In particular, researchers will need to explore how AI-driven decision support systems can improve treatment planning by evaluating patient-specific variables to identify the best therapeutic options [104]. Enhancing the interpretability and transparency of AI algorithms in pediatric urology should also be a priority, thereby building confidence and promoting cooperation between AI developers and healthcare practitioners [105].

Big data analytics and ML techniques also offer the possibility of identifying latent patterns, biomarkers, and predictive variables that lead to new understandings of pediatric urological disorders [106]. Standardization and validation will be essential to guarantee the robustness, dependability, and generalizability of AI models across populations and healthcare settings. The application of AI tools in pediatric urology has social, legal, and ethical ramifications that need to be considered in parallel. These include concerns about patient permission, data privacy, and fair access to AI-driven medical technology [104], which will undoubtedly lead to major improvements in clinical decision-making, patient care, and innovation in pediatric urology (Table 11).
diagnostics-14-02059-t011_Table 11Table 11Recommendations for future research and clinical practice.Recommendations Focus Area Action Item Conduct prospective studies [83]Long-term outcomes and cost effectiveness of AI-assisted surgical techniques in pediatric pyeloplasty Evaluate the long-term outcomes and cost effectiveness of AI-assisted surgical techniques in pediatric pyeloplasty. This includes assessing factors such as postoperative complications, recurrence rates, patient satisfaction, and economic implications over an extended follow-up period. Develop standardized protocols and guidelines [106]Ethical and responsible use of AI in pediatric urology, with a focus on patient privacy, data security, and algorithmic transparency Develop standardized protocols and guidelines for the ethical and responsible use of AI in pediatric urology. Emphasis should be placed on safeguarding patient privacy, ensuring robust data security measures, and promoting algorithmic transparency to maintain trust and integrity in AI-driven healthcare applications. Investigate the impact of AI-driven decision support systems [4]Impact of AI-driven decision support systems on clinical workflow, patient outcomes, and healthcare resource utilization in pediatric urology practice Investigate the impact of AI-driven decision support systems on various aspects of pediatric urology practice. This includes evaluating their influence on clinical workflow efficiency, patient outcomes such as surgical results and complication rates, as well as healthcare resource utilization including operative times, hospital length of stay, and overall healthcare costs. Additionally, explore user satisfaction and acceptance of these systems among healthcare providers. Explore applications of AI in preoperative planning and simulation [3]Optimization of surgical planning and training in pediatric urology using AI-driven simulation and modeling techniques Investigate the potential applications of AI in enhancing preoperative planning and simulation in pediatric urology, using AI algorithms to analyze patient-specific data and anatomical variations, thereby optimizing surgical strategies and facilitating personalized treatment plans. Additionally, explore the use of AI-driven virtual reality simulations for surgical training and skill enhancement among pediatric urologists. Assess the role of AI in predictive analytics and personalized medicine [5]Integration of AI-based predictive analytics for prognostication and personalized treatment approaches in pediatric urological conditions Assess the utility of AI-based predictive analytics in pediatric urology for predicting disease progression, treatment response, and individualized risk stratification. Explore the potential of AI algorithms in analyzing multidimensional datasets and biomarker profiles to tailor treatment strategies and optimize patient outcomes in pediatric urological conditions such as vesicoureteral reflux, hydronephrosis, and urinary tract infection. Investigate the application of AI for enhancing patient communication and engagement [2]Development of AI-powered tools to facilitate patient education, communication, and shared decision making in pediatric urology Investigate the use of AI-driven technologies, such as chatbots and virtual assistants, to enhance patient communication, education, and engagement in pediatric urology. Explore the potential of these tools to provide personalized information, support treatment adherence, and facilitate shared decision making among healthcare providers, patients, and their families. Evaluate the impact of AI on disparities in pediatric urological care [105]Examination of the potential influence of AI technologies on healthcare disparities and access to pediatric urological services Evaluate the impact of AI-driven technologies on disparities in pediatric urological care, including access to specialized services, diagnostic accuracy, and treatment outcomes among underserved populations. Identify potential barriers to and challenges facing equitable implementation of AI in pediatric urology and develop strategies to mitigate disparities and promote inclusive healthcare delivery. 


## Figures and Tables

**Figure 1 diagnostics-14-02059-f001:**
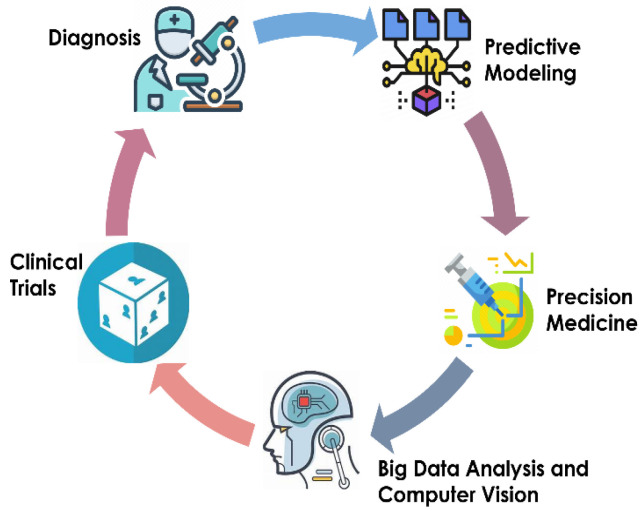
AI for the precise identification of pediatric urology conditions.

**Figure 2 diagnostics-14-02059-f002:**
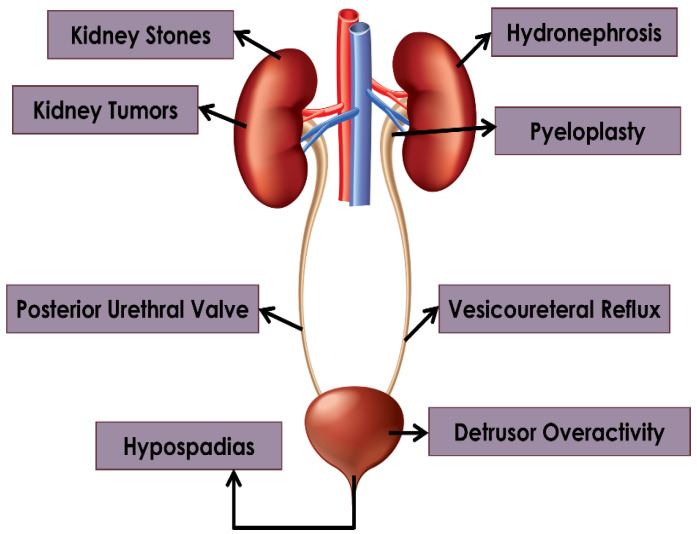
Pediatric urological disorders discussed in this review.

**Figure 8 diagnostics-14-02059-f008:**
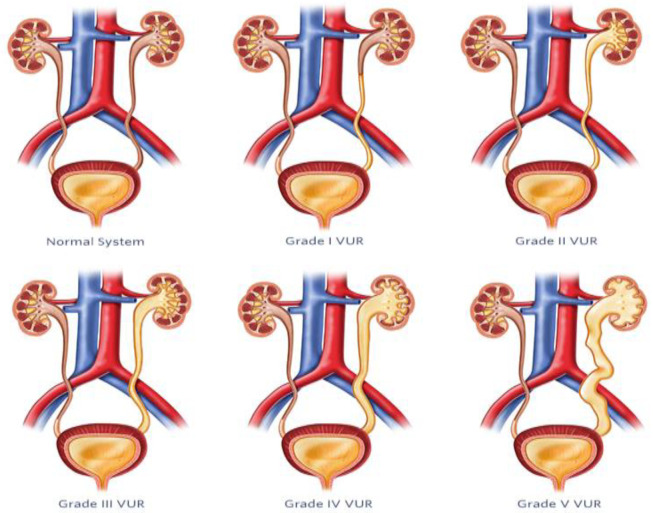
Vesicoureteral reflux (VUR) grading.

**Figure 9 diagnostics-14-02059-f009:**
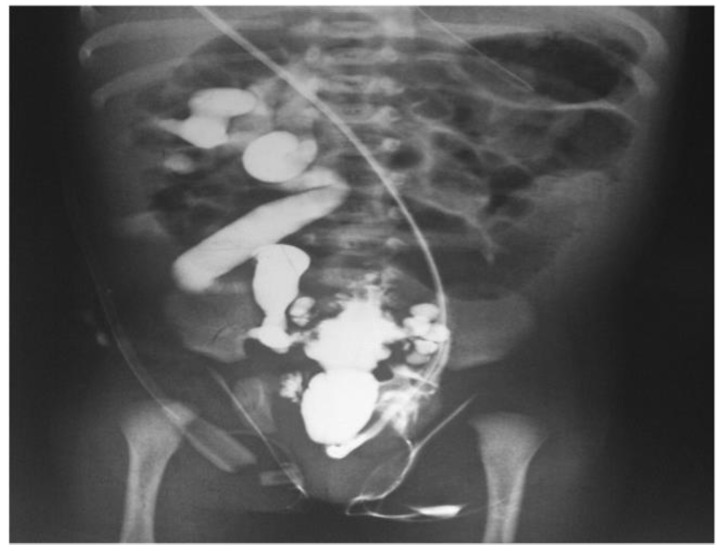
Micturating cystourethrogram revealing the presence of type 1 PUV together with corresponding alterations in the bladder (trabeculation) and upper urinary system (vesicoureteral reflux) [63].

## Data Availability

No new data were created or analyzed in this study. Data sharing is not applicable to this article.

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
