# Peer review of "Artificial Intelligence Tools in Pediatric Urology: A Comprehensive Review of Recent Advances"

_diagnostics, 2024, doi:10.3390/diagnostics14182059_

Round 1
Reviewer 1 Report
Comments and Suggestions for Authors
The article reviewed is a very detailed and extensive narrative analysis of a very up-to-date subject – Artificial Intelligence and specifically the place AI has in pediatric urology, a field where decision-making in the diagnosis, treatment and follow-up of a pathology will have a critical effect on the patient’s quality of life. The main purpose of the paper was to have a complete analysis of current state of AI integration in the medical care and research for pediatric patients with disorders of the urinary tract.
The authors have been very thorough, evaluating each of the most important urological disorders that pediatric urologists may encounter: hydronephrosis, urinary lithiasis, urinary tract tumors, ureteropelvic junction syndrome, vesicoureteral reflux, detrusor overactivity, posterior urethral valves, hypospadias but also evaluates research on other emerging AI applications.
As stated before, the main body of the paper is an in-extenso presentation of current research for each of the pediatric urological pathologies. A very high amount of work was put in determining what the current challenges are for each pathology, what is currently missing from the standpoint of either diagnostic, management or follow up and how can AI be implemented in order to mitigate those weak points in either guidelines, imaging limitations, treatment plans, surgical technique planning or long-term monitoring.
The conclusions were very clear and concise and determined the usefulness of AI for: enhancement of imaging evaluations for a better diagnosis, treatment planning and surgery results, increasing of diagnosis accuracy, personalized treatment planning, enhanced surgical guidance, increased productivity and enhanced education and training.
The authors were very careful to also include in the “Conclusions” chapter a subchapter on “Ethical and regulatory considerations”, because as we all know using AI in medical decision-making raises a lot of ethical and legal concerns in regard to accountability and responsibility of the medical act.
In the final chapters the authors presented recommendations for future research which may help with integration of AI systems in pediatric urological healthcare, but these points are universally true for all medical and surgical specialties: the need for more prospective studies, the necessity for developing standardized protocols and guidelines, the evaluation of the impact of AI driven decision support systems, exploration of applications of AI in surgery planning and simulation, assessment of the role of AI in predictive analysis and personalized medicine, and evaluation of the impact of AI on pediatric urological care as a whole.
It’s worth mentioning that the strong point of this paper – the fact that is very extensive also becomes a weak point, as the size of the paper might be a little exhaustive to read as a whole.
My conclusions are that this is a very well conceptualized paper, being clear and relevant for the current medical environment, bringing together the newest research on AI in pediatric urology bringing. The paper is very well organized, comprehensively describes the subject and presents research from highly praised authors. Even though the paper focuses on a very specific area of medical care, the conclusions may be extrapolated in other medical specialties and opens the doors for future research. The English language is very good, is easily understandable and appropriate. For these reasons I find that this work is of good quality, has a significant scientific contribution and is very well suited for the “Diagnostics” journal and I recommend the acceptance of the paper for publishing.
Author Response
Comments 1: [The article reviewed is a very detailed and extensive narrative analysis of a very up-to-date subject – Artificial Intelligence and specifically the place AI has in pediatric urology, a field where decision-making in the diagnosis, treatment and follow-up of a pathology will have a critical effect on the patient’s quality of life. The main purpose of the paper was to have a complete analysis of current state of AI integration in the medical care and research for pediatric patients with disorders of the urinary tract.
The authors have been very thorough, evaluating each of the most important urological disorders that pediatric urologists may encounter: hydronephrosis, urinary lithiasis, urinary tract tumors, ureteropelvic junction syndrome, vesicoureteral reflux, detrusor overactivity, posterior urethral valves, hypospadias but also evaluates research on other emerging AI applications.
As stated before, the main body of the paper is an in-extenso presentation of current research for each of the pediatric urological pathologies. A very high amount of work was put in determining what the current challenges are for each pathology, what is currently missing from the standpoint of either diagnostic, management or follow up and how can AI be implemented in order to mitigate those weak points in either guidelines, imaging limitations, treatment plans, surgical technique planning or long-term monitoring.
The conclusions were very clear and concise and determined the usefulness of AI for: enhancement of imaging evaluations for a better diagnosis, treatment planning and surgery results, increasing of diagnosis accuracy, personalized treatment planning, enhanced surgical guidance, increased productivity and enhanced education and training.
The authors were very careful to also include in the “Conclusions” chapter a subchapter on “Ethical and regulatory considerations”, because as we all know using AI in medical decision-making raises a lot of ethical and legal concerns in regard to accountability and responsibility of the medical act.
In the final chapters the authors presented recommendations for future research which may help with integration of AI systems in pediatric urological healthcare, but these points are universally true for all medical and surgical specialties: the need for more prospective studies, the necessity for developing standardized protocols and guidelines, the evaluation of the impact of AI driven decision support systems, exploration of applications of AI in surgery planning and simulation, assessment of the role of AI in predictive analysis and personalized medicine, and evaluation of the impact of AI on pediatric urological care as a whole.
It’s worth mentioning that the strong point of this paper – the fact that is very extensive also becomes a weak point, as the size of the paper might be a little exhaustive to read as a whole.
My conclusions are that this is a very well conceptualized paper, being clear and relevant for the current medical environment, bringing together the newest research on AI in pediatric urology bringing. The paper is very well organized, comprehensively describes the subject and presents research from highly praised authors. Even though the paper focuses on a very specific area of medical care, the conclusions may be extrapolated in other medical specialties and opens the doors for future research. The English language is very good, is easily understandable and appropriate. For these reasons I find that this work is of good quality, has a significant scientific contribution and is very well suited for the “Diagnostics” journal and I recommend the acceptance of the paper for publishing.]
Response 1: [Thank you very much for your detailed and positive review of our manuscript titled. We greatly appreciate your commendations on the thoroughness, organization, and scientific contribution of our work.
We acknowledge your observation regarding the length of the paper and understand that it might be perceived as exhaustive. The comprehensive nature of this review is intentional, as we aimed to provide a complete and detailed analysis of the current state of AI integration in pediatric urology. Given the complexity of the field and the rapid advancements in AI technologies, covering a wide range of disorders and emerging applications requires a detailed examination to ensure that all relevant aspects are addressed.
Our goal was to offer an in-depth resource that would be valuable by presenting a thorough overview of AI applications in pediatric urology, including challenges, knowledge gaps, and future directions. We believe that the length of the paper is necessary to fulfill this objective and to provide a comprehensive reference for ongoing and future research.
We appreciate your understanding of the need for this detailed approach and are pleased that you found the manuscript to be clear, relevant, and of significant scientific contribution.
Thank you again for your valuable feedback and for recommending the acceptance of our paper for publication.]
Reviewer 2 Report
Comments and Suggestions for Authors
Regarding the study with the title "Artificial intelligence tools in pediatric urology: a comprehensive review of recent advances"
In which reviewed the applications of learning artificial intelligence in the field of pediatric urology. I would like to inform the authors that this manuscript has been done completely and comprehensively. Its methodology is also suitable and without problems. It seems that it can be published with minimal revisions. But there are some very small points that by considering these points, the quality of the study can be greatly improved.
The points to consider are as follows
1: The abstract desperately needs to be enriched. Mention the findings and a summary of the findings in the abstract. The abstract you wrote is without a doubt soulless.
2: Be sure to add a section in the introduction as artificial intelligence and its applications in the field of health care. This part can make the introduction more complete and greatly increase the readability of this study. For this purpose, you can refer to the following studies that were inspired by artificial intelligence in other fields.
"Deep learning model utilization for mortality prediction in mechanically ventilated ICU patients"
"A Multimodal Intermediate Fusion Network with Manifold Learning for Stress Detection"
The use of these studies can greatly expand the authors' view to more applications of artificial intelligence in the field of health care.
The rest of the manuscript is well done. A complete discussion has taken place. The content provided is appropriate and complete. By doing these revisions, it can be highly recommended for publication.
Author Response
Thank you very much for your thoughtful feedback on our manuscript. We appreciate your positive evaluation of the study and are grateful for your constructive suggestions. Below, we address each of your comments:
Comment 1: [The abstract desperately needs to be enriched. Mention the findings and a summary of the findings in the abstract. The abstract you wrote is without a doubt soulless.]
Response 1: Thank you for highlighting the need to enhance the abstract. We have revised it to better reflect the findings and key results of our study. The updated abstract now includes a summary of the significant impacts of AI on diagnostic accuracy, surgical precision, personalized treatment planning, and advancements in imaging techniques such as enhancement and segmentation. We believe these additions address the need for a more engaging and informative abstract.
Comment 2: [Be sure to add a section in the introduction as artificial intelligence and its applications in the field of health care. This part can make the introduction more complete and greatly increase the readability of this study. For this purpose, you can refer to the following studies that were inspired by artificial intelligence in other fields: ‘Deep learning model utilization for mortality prediction in mechanically ventilated ICU patients’ and ‘A Multimodal Intermediate Fusion Network with Manifold Learning for Stress Detection.’ The use of these studies can greatly expand the authors' view to more applications of artificial intelligence in the field of health care.]
Response 2: We appreciate your recommendation to broaden the introduction with a discussion of AI applications in healthcare. In response, we have added a new section titled "AI Applications in Other Medical Fields" to the introduction. This section includes the studies you suggested and also incorporates additional relevant research on AI technologies for diagnosing aortitis and predicting clinical outcomes in conditions such as paralytic ileus and diabetes mellitus. This addition is designed to provide a more comprehensive background and enhance the readability of the manuscript. The new content is located on pages 4 and 5 of the introduction, with the added text highlighted for your convenience.
Thank you again for your valuable feedback and for recommending our paper for publication. We hope the revisions meet your expectations and further improve the quality of the manuscript.